# Spiciness theory revisited, with new views on neutral density, orthogonality and passiveness

Rémi Tailleux[1]

[1]Dept of Meteorology, University of Reading, Earley Gate, PO Box 243, RG6 6BB Reading, United Kingdom

**Correspondence:** Rémi Tailleux (R.G.J.Tailleux@reading.ac.uk)

**Abstract.** This paper clarifies the theoretical basis for constructing spiciness variables optimal for characterising ocean water masses. Three essential ingredients are identified: 1) a material density variable $\gamma$ that is as neutral as feasible; 2) a material state function $\xi$ independent of $\gamma$, but otherwise arbitrary; 3) an empirically determined reference function $\xi_r(\gamma)$ of $\gamma$ representing the imagined behaviour of $\xi$ in a notional spiceless ocean. Ingredient 1) is required because contrary to what is often assumed, it is not the properties imposed on $\xi$ (such as orthogonality) that determine its dynamical inertness but the degree of neutrality of $\gamma$. The first key result is that it is the anomaly $\xi' = \xi - \xi_r(\gamma)$, rather than $\xi$, that is the variable the most suited for characterising ocean water masses, as originally proposed by McDougall and Giles (1987). The second key result is that oceanic sections of normalised $\xi'$ appear to be relatively insensitive to the choice of $\xi$, as first suggested by Jackett and McDougall (1985), based on the comparison of very different choices of $\xi$. It is also argued that orthogonality of $\nabla \xi'$ to $\nabla \gamma$ in physical space is more germane to spiciness theory than orthogonality in thermohaline space, although how to use it to constrain the choices of $\xi$ and $\xi_r(\gamma)$ remains to be fully elucidated. The results are important for they unify the various ways in which spiciness has been defined and used in the literature. They also provide a rigorous theoretical basis justifying the pursuit of a globally defined material density variable maximising neutrality. To illustrate the latter point, this paper proposes a new implementation of the author's recently developed thermodynamic neutral density and explains how to adapt existing definitions of spiciness/spicity to work with it.

## 1 Introduction

As is well known, three independent variables are needed to fully characterise the thermodynamic state of a fluid parcel in the standard approximation of seawater as a binary fluid. The standard description usually relies on the use of a temperature variable (such as potential temperature $\theta$, in-situ temperature $T$ or Conservative Temperature $\Theta$), a salinity variable (such as reference composition salinity $S$ or Absolute Salinity $S_A$), and pressure $p$. In contrast, theoretical descriptions of oceanic motions only require the use of two 'active' variables, namely in-situ density $\rho$ and pressure. The implication is that $S$ and $\theta$ can be regarded as being made of an 'active' part contributing to density, and a passive part associated with density-compensated variations in

$\theta$ and $S$ — usually termed 'spiciness' anomalies — which behaves as a passive tracer. Physically, such an idea is empirically supported by numerical simulation results showing that the turbulence spectra of density-compensated thermohaline variance is generally significantly different from that contributing to the density (Smith and Ferrari, 2009).

Although behaving predominantly as passive tracers, density-compensated anomalies may however occasionally 'activate' and couple with density and ocean dynamics. This may happen, for instance, when isopycnal mixing of $\theta$ and $S$ leads to cabelling and densification, which may create available potential energy (Butler et al., 2013); when density-compensated temperature anomalies propagate over long distances to de-compensate upon reaching the ocean surface, thus modulating air-sea interactions (Lazar et al., 2001); when density-compensated salinity anomalies propagate from the equatorial regions to the regions of deep water formation, thus possibly modulating the strength of the thermohaline circulation (Laurian et al., 2006, 2009); when isopycnal stirring of density-compensated $\theta/S$ anomalies releases available potential energy associated with thermobaric instability (Ingersoll, 2005; Tailleux, 2016a). For these reasons, the mechanisms responsible for the formation, propagation, and decay of spiciness anomalies have received much attention, with a key research aim being to understand their impacts on the climate system, e.g., Schneider (2000), Yeager and Large (2004), Luo et al. (2005), Tailleux et al. (2005), Zika et al. (2020).

From a dynamical viewpoint, in-situ density $\rho$ is the most relevant density variable for defining density-compensated $\theta/S$ anomalies but its strong pressure dependence makes the associated isopycnal surfaces strongly time dependent and therefore impractical to use. This is why in practice oceanographers prefer to work with isopycnal surfaces defined by means of a purely material density-like variable $\gamma = \gamma(S, \theta)$ unaffected by pressure variations. Since density-compensated $\theta/S$ anomalies are truly passive only if defined in terms of in-situ density, $\gamma$ needs to be able to mimic the dynamical properties of in-situ density as much as feasible. As discussed by Eden and Willebrand (1999), this amounts to imposing that $\gamma$ be constructed to be as neutral as feasible. Because of the thermobaric nonlinearity of the equation of state, it is well known that exact neutrality cannot be achieved by any material variable. As a result, investigators have resorted to using either neutral surfaces (McDougall and Giles, 1987) or potential density referenced to a pressure close to the range of pressures of interest (Jackett and McDougall, 1985; Huang, 2011; McDougall and Krzysik, 2015; Huang et al., 2018). In this paper, I propose instead to use a new implementation of Tailleux (2016b)'s thermodynamic neutral density variable $\gamma^T$, which is currently the most neutral material density-like variable available. The resulting new variable is referred to as $\gamma^T_{analytic}$ in the following and details of its construction and implementation are given in Section 2.

Once a choice for $\gamma$ has been made, a second material variable $\xi = \xi(S, \theta)$ is required to fully characterise the thermodynamic properties of a fluid parcel. From a mathematical viewpoint, the only real constraint on $\xi$ is that the transformation $(S, \theta) \to (\gamma, \xi)$ defines a continuously differentiable one-to-one mapping (that is, an isomorphism) so that $(S, \theta)$ properties can be recovered from the knowledge of $(\gamma, \xi)$. For this, it is sufficient that the Jacobian $J = \partial(\xi, \gamma)/\partial(S, \theta)$ differs from zero everywhere in $(S, \theta)$ space where invertibility is required. Historically, however, spiciness theory appears to have been developed on the predicate that for $\xi$ to be dynamically inert, it should be constructed to be 'orthogonal' to $\gamma$ in $(S, \theta)$ space, as originally put forward by Veronis (1972) (whose variable is denoted by $\tau^\nu$ in the following). This notion was argued to be incorrect by Jackett and McDougall (1985), however, who pointed out: "[...] the variations of any variable, when measured along

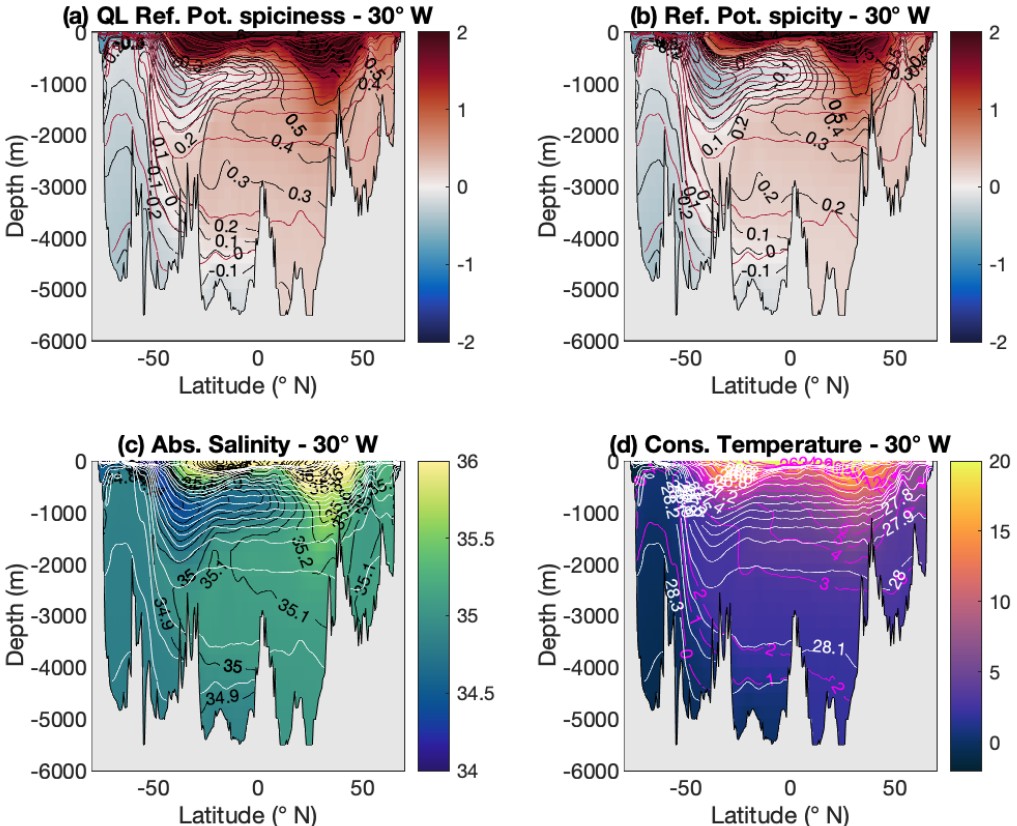

**Figure 1.** Comparison of different spiciness-as-state-functions along $30°W$ in the Atlantic ocean: (a) a new form of potential spiciness $\tau_{ref} = \tau_{\ddagger}(S_A, \Theta, p_r)$ referenced to a variable reference pressure $p_r(S, \theta)$. This spiciness variable is similar to McDougall and Krzysik (2015) spiciness variable and defined in Section 3. The variable reference pressure $p_r$ is defined in Section 2 and illustrated in panel (a) of Fig. 3. (b) Huang et al. (2018) potential spicity referenced to the same variable reference pressure $p_r$ as in (a), denoted by $\pi_{ref}$ in the paper; (c) Absolute Salinity; (d) Conservative Temperature. White contours in panels (c) and (d) (shown as brown contours in panels (a) and (b)) represent selected isocontours of a density-like variable $\gamma^T_{analytic}$ similar to Tailleux (2016b)'s thermodynamic neutral density variable $\gamma^T$. The construction and implementation of $\gamma^T_{analytic}$ are described in Section 2. These isocontours — the same in all panels — are only labelled in panel (d) for clarity.

isopycnal surfaces, are dynamically passive and so the perpendicular property does not, of itself, contribute to the dynamic
inertness of $\tau^\nu$", to which they added: "Secondly, it is readily apparent that the perpendicular property itself has no inherent physical meaning since a simple rescaling of either the potential temperature $\theta$ or the salinity axis $S$ destroys the perpendicular property." Jackett and McDougall (1985)'s remarks are important for at least two reasons: 1) for suggesting that it is really the isopynal anomaly $\xi' = \xi - \xi_r(\gamma)$ defined relative to some reference function of density $\xi_r(\gamma)$ that is dynamically passive

and therefore the quantity truly measuring spiciness. This was further supported by McDougall and Giles (1987) subsequently arguing that it is such an anomaly that represents the most appropriate approach for characterising water mass intrusions. Note here that if one defines reference salinity and temperature profiles $S_r(\gamma)$ and $\theta_r(\gamma)$ such that $\gamma(S_r(\gamma_0), \theta_r(\gamma_0)) = \gamma_0$, using a Taylor series expansion shows that $S' = S - S_r(\gamma)$ and $\theta' = \theta - \theta_r(\gamma)$ are $\gamma$-compensated at leading order, i.e., they satisfy $\gamma_S S' + \gamma_\theta \theta' \approx 0$, and hence approximately passive; 2) for establishing that the imposition of any form of orthogonality between $\xi$ and $\gamma$ is even less meaningful than previously realised, since that even if $\xi$ is constructed to be orthogonal to $\gamma$ in some sense, this orthogonality is lost by the anomaly $\xi' = \xi - \xi_r(\gamma)$.

If one accepts that it is the spiciness anomaly $\xi' = \xi - \xi_r(\gamma)$ rather than the spiciness-as-a-state-function $\xi$ that is the most appropriate measure of water mass contrasts, as argued by Jackett and McDougall (1985) and McDougall and Giles (1987), the central questions that the theory of spiciness needs to address become:

1. Are there any special benefits associated with one particular choice of spiciness-as-a-state-function $\xi$ over another, and if so, what are the relevant physical arguments that should be invoked to establish the superiority of any given particular choice of $\xi$?

2. How should $\gamma$ and the reference function $\xi_r(\gamma)$ be constructed and justified?

3. While any $\xi$ independent of $\gamma$ can be used to meaningfully compare the spiciness of two different water samples lying on the same isopycnal surface $\gamma = \text{constant}$, it is generally assumed that it is not possible to meaningfully compare the spiciness of two water samples belonging to two different isopycnal surfaces $\gamma_1$ and $\gamma_2$ (Timmermans and Jayne, 2016). Is this belief justified? Isn't the transformation of $\xi$ into an anomaly $\xi'$ sufficient to address the issue?

Anent the first question, Jackett and McDougall (1985) have developed geometrical arguments in support of a variable $\tau_{jmd}$ satisfying $\int d\tau_{jmd} = \int \beta dS$ along potential density surfaces that they argue make it superior to other choices. These arguments do not seem to be decisive, however, since $\tau_{jmd}$ does not satisfy the above mentioned invertibility constraint where the thermal expansion $\alpha$ vanishes. At such points, temperature becomes approximately passive and therefore the most natural definition of spiciness as pointed out by Stipa (2002). At such points, however, Jackett and McDougall (1985)'s variable, like Flament (2002)'s variable, behave like salinity, causing the Jacobian of the transformation $\partial(\tau_{jmd}, \gamma)/\partial(S, \theta)$ to vanish, which seems unphysical. The ability of different kinds of anomalies, namely $\tau'_\nu$, $\tau'_{jmd}$ and $S'$, to characterise water mass contrasts and intrusions is discussed in the second part of Jackett and McDougall (1985)'s paper. Interestingly, they find that even though the spiciness-as-state-functions $\tau^\nu$, $\tau_{jmd}$ and $S$ behave quite differently from each other in $(S, \theta)$ space, their anomalies exhibit in contrast only small differences, at least when estimated for individual soundings. In this paper, I show that this property actually appears to be satisfied much more broadly, as illustrated in Fig. 12 and further discussed in the text.

Orthogonality in $(S, \theta)$ space — despite its usefulness or necessity remaining a source of confusion and controversy — has nevertheless been central to the development of spiciness theory. Recently, Huang et al. (2018) attempted to rehabilitate Veronis (1972)'s form of orthogonality by arguing that without imposing it, it is otherwise hard to define a distance in $(S, \theta)$ space. This argument is unconvincing, however, because the concept of distance in mathematics does not require orthogonality, it only

requires the introduction of a positive definite metric $d(x, y)$, i.e., one satisfying: 1) $d(x, y) \geq 0$ for all x and y; 2) $d(x, y) = 0$ is equivalent to $x = y$; 3) $d(x, y) = d(y, x)$; 4) $d(x, y) \leq d(x, z) + d(z, y)$, the so-called triangle inequality. As a result, there is an infinite number of ways to define distance in $(S, \theta)$ space. For instance, $d(A, B) = \sqrt{\beta_0^2 (S_A - S_B)^2 + \alpha_0^2 (\theta_A - \theta_B)^2}$, where $\alpha_0$ and $\beta_0$ are some constant reference values of $\alpha$ and $\beta$, is an acceptable definition of distance. Likewise, any two non-trivial and independent material functions $\gamma(S, \theta)$ and $\xi(S, \theta)$ could also be used to define $d(A, B) = \sqrt{(\gamma_A - \gamma_B)^2 + K_0^2 (\xi_A - \xi_B)^2}$, where $K_0$ is a constant to express $\gamma$ and $\xi$ in the same system of units if needed, while $\gamma_A$ is shorthand for $\gamma(S_A, \theta_A)$, with similar definitions for $\gamma_B$, $\xi_A$ and $\xi_B$. As regards to the 45 degrees orthogonality proposed by Jackett and McDougall (1997) and Flament (2002), while it is true that it is unaffected by a re-scaling of the $S$ and $\theta$ axes plaguing Veronis (1972) form of orthogonality, it is however destroyed by the subtraction of any function of potential density, while also causing the above mentioned loss of invertibility where $\alpha$ vanishes. In any case, it is unclear why Jackett and McDougall (1997) sought to impose the 45 degrees orthogonality to $\tau_{jmd}$, since it is a priori not necessary to satisfy the above-mentioned constraint $\int d\tau_{jmd} = \int \beta \, dS$ on isopycnal surfaces (in the sense that if one particular $\tau_{jmd}$ solves the problem, any $\tau_{jmd} \to \tau_{jmd} - \tau_r(\gamma)$ will also solve it).

From a purely empirical viewpoint, neither spiciness (however defined) nor Huang et al. (2018) spicity appears to have any particular advantage over salinity at picking up ocean water mass signals, although both variables are superior than temperature in this respect. This is shown in Fig. 1, which compares the aptitude of (a) reference potential spiciness $\tau_{ref}$, (b) reference potential spicity $\pi_{ref}$, (c) Absolute Salinity and (d) Conservative Temperature for visualising the water masses of the Atlantic Ocean along the $30°W$ section, of which the 4 main ones are North-Atlantic Deep Water (NADW), Antarctic Intermediate Water (AAIW), Antarctic Bottom Water (AABW) and Mediterranean Intermediate Water (MIW). The WOCE climatological dataset (Gouretski and Koltermann, 2004) [1] has been used for this figure as well as for all calculations throughout this paper. The link between $\tau_{ref}$ and $\pi_{ref}$ and the spiciness/spicity variables of McDougall and Krzysik (2015) and Huang et al. (2018) is explained in the next two sections. Fig. 1 shows that while all variables are able to pick up the AABW signal similarly well, they differ in their ability to pick up the AAIW signal. Indeed, AAIW is most prominently displayed in the salinity field, followed by $\pi_{ref}$ and $\tau_{ref}$, with Conservative Temperature a distant last, based on how well defined the signal is and how far north it can be tracked. As regards to NADW and MIW, they can be clearly identified in all variables but temperature.

Since salinity satisfies neither form of orthogonality, it is natural to ask which properties make it superior to spiciness and spicity as a water mass indicator? Because two signals are visually most easily contrasted when their respective isocontours are orthogonal to each other, it is natural to ask whether salinity could owe its superiority to being on average more orthogonal to density than other variables in *physical space*. To test this, the median angle between $\nabla \gamma$ and $\nabla \xi$ estimated for all available points of the WOCE dataset was chosen as an orthogonality metric and computed for the 4 spiciness-as-state-functions $\xi$ considered. The result is displayed in Fig. 2 (represented by the blue bars), both for the global ocean (top panel) as well for the $30°W$ Atlantic section only (bottom panel). The median angle between $\nabla \gamma$ and $\nabla \xi$ was favoured over other metrics of orthogonality owing to its ability to rank the spiciness-as-state-functions in the same order as the subjective visual determination

---

[1] available at:

http://icdc.cen.uni-hamburg.de/1/daten/index.php?id=woce&L=1

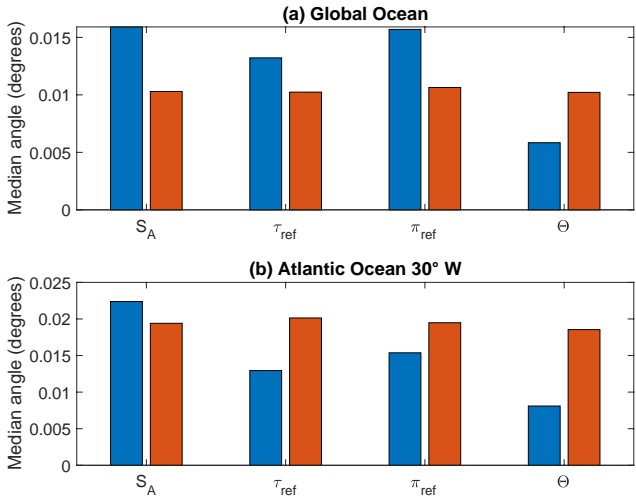

**Figure 2.** Median angle between $\nabla\gamma_{analytic}^T$ and $\nabla\xi$ (blue bars) as well as between $\nabla\gamma_{analytic}^T$ and $\nabla\xi'$ (red bars) for $\xi = S_A$, $\xi = \tau_{ref}$, $\xi = \pi_{ref}$ and $\xi = \Theta$. The top panel takes into account all available data points of the WOCE dataset, whereas the bottom panel only account for the data points making up the $30^\circ W$ Atlantic Ocean section. See Section 4 for details of the construction of $\xi'$. While the orthogonality to density of spiciness-as-state-functions depends sensitively on the variable considered (blue bars), this is much less the case of the corresponding spiciness-as-anomalies, as seen by the similar magnitude exhibited by the red bars.

of their ability as water mass indicators based on Fig. 1, with salinity first, $\pi_{ref}$ second, $\tau_{ref}$ third and temperature last. In this paper, this idea will be further explored by investigating whether the generally observed superior ability of $\xi'$ over $\xi$ as water mass indicators can be attributed to their increased orthogonality to $\gamma$. (The results turn out to be inconclusive).

    The main aim of this paper is to explore the above ideas further and to clarify their inter-linkages. One of its key points is to emphasise that spiciness is a property, not a substance, and hence that it is spiciness-as-an-anomaly rather than spiciness-as-

a-state function that is the relevant concept to quantify water mass contrasts. This important point was recognised early on by Jackett and McDougall (1985) and McDougall and Giles (1987) but for some reason has since been systematically overlooked in most of the recent spiciness theory literature devoted to the construction of dedicated spiciness-as-a-state-function variables, e.g., Flament (2002); McDougall and Krzysik (2015); Huang (2011); Huang et al. (2018). This is problematic, because this has resulted in a disconnect between spiciness theory and its applications. Section 2 emphasises the result that it is the degree of

neutrality of the density variable $\gamma$ serving to define density-compensation that determines the degree of dynamical inertness of spiciness variables $\xi$, not the properties of $\xi$ itself. Because Tailleux (2016b)'s thermodynamic neutral density $\gamma^T$ is currently the most neutral material density-like variable available, a new implementation of it, denoted $\gamma_{analytic}^T$, is proposed for use in spiciness studies. In contrast to $\gamma^T$, $\gamma_{analytic}^T$ can be estimated with only a few lines of code and is therefore much simpler to use in practice, while also being smoother and somewhat more neutral than $\gamma^T$. Because $\gamma_{analytic}^T$ relies on the use of

a non-constant reference pressure field $p_r(S,\theta)$, it can only be used in conjunction with spiciness-as-state-functions whose dependence on pressure is sufficiently detailed. While this is the case of Huang et al. (2018)'s spicity variable, this is not the

case of McDougall and Krzysik (2015)'s spiciness variable, which is only defined for 3 discrete reference pressures, namely $0\,\mathrm{dbar}$, $1000\,\mathrm{dbar}$ and $2000\,\mathrm{dbar}$. To remedy this problem, Section 3 discusses the construction of a mathematically explicit spiciness variable $\tau_{\ddagger}(S,\theta,p)$, which closely mimics the behaviour of McDougall and Krzysik (2015)'s variable in most of $(S,\theta)$

space. As a result, two new potential spiciness and spicity variables referenced to the non-constant reference pressure $p_r(S,\theta)$, denoted by $\tau_{ref}$ and $\pi_{ref}$ respectively, are introduced in this paper. These are the variables depicted in panels (a) and (b) of Fig. 1. Section 4 discusses the links between the zero of spiciness, the definition of a notional spiceless ocean, the construction of spiciness anomalies, the orthogonality to density in physical space, and whether it is possible to meaningfully compare the spiciness of two water samples that do not belong to the same density surface. Finally, Section 5 summarises the results and

discusses their implications and any further work needed.

## 2    On the choice of isopycnal surfaces for spiciness studies

### 2.1    Dynamical inertness of spiciness and neutrality

As mentioned above, density-compensated thermohaline variations are truly passive only if defined along surfaces of constant in-situ $\rho$. However, because such surfaces are sensitive to pressure variations, it is useful in practice to seek for a purely

material proxy $\gamma = \gamma(S,\theta)$ of in-situ density, which can also be used to capture the 'active' part of $S$ and $\theta$. By introducing an as yet undetermined additional spiciness-as-state-function $\xi$ to capture the 'passive' part of $S$ and $\theta$, in-situ density may thus be rewritten as a function of the new $(\gamma,\xi,p)$ coordinates as $\rho = \rho(S,\theta,p) = \hat{\rho}(\gamma,\xi,p)$, a hat being used for the $(\gamma,\xi,p)$ representation of any function of $(S,\theta,p)$. By using the so-called Jacobi method, e.g., see Appendix A of Feistel (2018), Tailleux (2016a) showed that the partial derivatives of $\hat{\rho}$ with respect to $\gamma$ and $\xi$ are given by:

$$\frac{\partial \hat{\rho}}{\partial \gamma} = \frac{\partial(\hat{\rho},\xi)}{\partial(\gamma,\xi)} = \frac{1}{J}\frac{\partial(\xi,\rho)}{\partial(S,\theta)} = \frac{J_{\gamma}}{J}, \tag{1}$$

$$\frac{\partial \hat{\rho}}{\partial \xi} = \frac{\partial(\gamma,\hat{\rho})}{\partial(\gamma,\xi)} = \frac{1}{J}\frac{\partial(\rho,\gamma)}{\partial(S,\theta)} = \frac{J_{\xi}}{J}, \tag{2}$$

where $J_{\gamma} = \partial(\xi,\rho)/\partial(S,\theta)$, $J_{\xi} = \partial(\gamma,\rho)/\partial(S,\theta)$ and $J = \partial(\xi,\gamma)/\partial(S,\theta)$. To clarify the conditions controlling the passive character of $\xi$, it is useful to derive the following expression for the neutral vector $\mathbf{N}$ in $(\gamma,\xi,p)$ coordinates:

$$\mathbf{N} = -\frac{g}{\hat{\rho}}\left(\nabla\hat{\rho} - \hat{\rho}_p\nabla p\right) = -\frac{g}{\hat{\rho}}\left(\hat{\rho}_{\gamma}\nabla\gamma + \hat{\rho}_{\xi}\nabla\xi\right), \tag{3}$$

where $\hat{\rho}_p = \partial\hat{\rho}/\partial p$, $\hat{\rho}_{\gamma} = \partial\hat{\rho}/\partial\gamma$, $\hat{\rho}_{\xi} = \partial\hat{\rho}/\partial\xi$. Because the Jacobian $J$ is invariant upon the transformation $\xi \to \xi - \xi_r(\gamma)$, the expression for $\mathbf{N}$ may alternatively be written in terms of $\nabla\gamma$ and $\nabla\xi' = \nabla(\xi - \xi_r(\gamma))$ as follows:

$$\mathbf{N} = -\frac{g}{\rho}\left[\left(\hat{\rho}_{\gamma} + \hat{\rho}_{\xi}\frac{d\xi_r}{d\gamma}\right)\nabla\gamma + \hat{\rho}_{\xi}\nabla\xi'\right]. \tag{4}$$

Physically, the condition for $\xi$ or $\xi'$ to be dynamically inert is that it does not affect $\hat{\rho}$, which mathematically requires $\hat{\rho}_{\xi} = 0$.

Eqs. (3) or (4) show this is equivalent to $\gamma$ being exactly neutral. Eq. (2) shows that this can only be the case if $\partial(\rho,\gamma)/\partial(S,\theta) =$

0. As is well known, this condition can never be completely satisfied in practice because of thermobaricity, i.e., the pressure dependence of the thermal expansion coefficient (McDougall 1987; Tailleux (2016a)). The above conditions thus establish that the degree of dynamical inertness of $\xi$ is controlled by the degree of non-neutrality of $\gamma$. This is an important result for two reasons. First, because it shows that the degree of dynamical inertness of $\xi$ is not determined its properties (such as orthogonality) but by those of $\gamma$. Second, because it provides new rigorous theoretical arguments for justifying the pursuit of a globally defined material density variable $\gamma(S,\theta)$ maximising neutrality (although this won't come as a surprise to most oceanographers).

## 2.2 A new implementation of thermodynamic neutral density for spiciness studies and water mass analyses

Until recently, isopycnal analysis in oceanography has relied on two main approaches: the use of vertically stacked potential densities referenced to a discrete set of reference pressures 'patched' at the points of discontinuity following Reid (1994), a.k.a. patched potential density (PPD), and the use of empirical neutral density $\gamma^n$ proposed as a continuous analogue of patched potential density by Jackett and McDougall (1997). Neither variable is exactly material, however. For PPD, this is because the points of discontinuities at which potential density referenced to different reference pressures are a source of non-materiality, see deSzoeke and Springer (2009). For $\gamma^n$, this is because such a variable also depends on horizontal position and pressure, although a way to remove the pressure dependence was recently proposed Lang et al. (2020).

Recently, Tailleux (2016b) pointed out that Lorenz reference density $\rho_{LZ}(S,\theta) = \rho(S,\theta,p_r(S,\theta))$ that enters Lorenz (1955) theory of available potential energy (APE) (see Tailleux (2013b) for a review) could be viewed as a generalisation of the concept of potential density referenced to the pressure $p_r(S,\theta)$ that a parcel would have in a notional state of rest. A computationally efficient approach to estimate the reference density and pressure vertical profiles $\rho_0(z)$ and $p_0(z)$ that characterise Lorenz reference state was proposed by Saenz et al. (2015). Once the latter are known, the reference pressure $p_r = p_0(z_r)$ is simply obtained by solving Tailleux (2013a)'s Level Neutral Buoyancy (LNB) equation

$$\rho(S,\theta,p_0(z_r)) = \rho_0(z_r) \tag{5}$$

for the reference depth $z_r$. As it turns out, $\rho_{LZ}$ happens to be quite neutral away from the polar regions where fluid parcels are close to their reference position. However, like in-situ density, $\rho_{LZ}$ is dominated by compressibility and its dependence on pressure. Tailleux (2016b) defined the thermodynamic neutral density variable

$$\gamma^T = \rho(S,\theta,p_r) - f(p_r) \tag{6}$$

as a modified form of Lorenz reference density empirically corrected for pressure and realised that the empirical correction function $f(p_r)$ could be chosen so that $\gamma^T$ closely approximates Jackett and McDougall (1997) empirical neutral density $\gamma^n$ outside the ACC [2].

Thermodynamic neutral density $\gamma^T$ is attractive because it is as far as we know the most neutral purely material density-like variable around. Unlike PPD, it varies smoothly and continuously across all pressure ranges. Moreover, it also provides a non-

---

[2]The software used to compute $\gamma^n$ was obtained from the TEOS-10 website at http://www.teos-10.org/preteos10_software/neutral_density.html

constant reference pressure $p_r(S, \theta)$ that can serve to define potential spiciness and potential spicity variables possessing the same degree of smoothness as $\gamma^T$, whose construction is discussed in Section 3. At present, $\gamma^T$ is therefore the most natural choice for use in spiciness studies since $\gamma^n$, which although somewhat more neutral, is not purely material. However, Tailleux (2013b)'s original implementation of $\gamma^T$ is a multi-step process starting with the computation of Lorenz reference state, which although made relatively easy by Saenz et al. (2015)'s method, is computationally involved and therefore not necessarily easily reproducible by others. To circumvent these difficulties, I am proposing here an alternative construction of $\gamma^T$, called $\gamma^T_{analytic}$, which by contrast is easily computed with only a few lines of code, while being also smoother and somewhat more neutral than $\gamma^T$. The proposed approach relies on using analytic profiles for $\rho_0(z)$ and $p_0(z)$ instead of determining these empirically, given by:

$$\rho_0(z) = \frac{a(z+e)^{b+1}}{b+1} + cz + d, \tag{7}$$

$$p_0(z) = g \left[ \frac{a(z+e)^{b+2}}{(b+1)(b+2)} + \frac{cz^2}{2} + dz - \frac{ae^{b+2}}{(b+1)(b+2)} \right], \tag{8}$$

where $z$ is positive depth increasing downward. The reference density profile $\rho_0(z)$ depends on 5 parameters $(a, b, c, d, e)$, which were estimated by fitting the top-down Lorenz reference density profile of Saenz et al. (2015). The results of the fitting procedure are given in Table A1. As to the empirical pressure correction $f(p)$ entering (6), it is chosen as a polynomial of degree 9 of the normalised pressure $(p - p_m)/\Delta p$ and given by the following expression:

$$f(p) = \sum_{n=1}^{9} a_n \left( \frac{p - p_m}{\Delta p} \right)^{9-n}. \tag{9}$$

The values of the coefficients $a_n$, $p_m$, $\Delta p$, and estimates of confidence intervals returned by the fitting procedure are given in Table A2.

A full account of the performances and properties of $\gamma^T_{analytic}$ will be reported in a forthcoming paper and is therefore outside the scope of this paper. Here, I only show two illustrations that are sufficient to justify the usefulness of $\gamma^T_{analytic}$ for the present purposes. Thus, the top panel of Fig. 3 depicts the reference pressure field $p_r(S, \theta) = p_0(z_r)$ along $30°W$ in the Atlantic ocean, while the bottom panel demonstrates the very close agreement between $\gamma^T$ and $\gamma^n$ outside the Southern Ocean along the same section (the calculations presented actually make use of the new thermodynamic standard, see Pawlowicz et al. (2012); IOC et al. (2010), and use Absolute Salinity $S_A$ and Conservative Temperature $\Theta$). Similarly good agreement was also verified in other parts of the ocean (not shown). Fig. 4 depicts latitude/depth sections along $30°W$ in the Atlantic ocean of the effective diffusivity $K_f = K_i \sin^2 (\mathbf{N}, \nabla\gamma)$ introduced by Hochet et al. (2019), a metric for the degree of non-neutrality similar to the concept of fictitious diffusivity used by Lang et al. (2020) and others, for: (a) $\gamma^n$, (b) $\gamma^T_{analytic}$, (c) $\sigma_1$ and (d) $\sigma_2$. As before, $\mathbf{N}$ is the neutral vector, $\gamma$ the density-like variable of interest, and $(\mathbf{N}, \nabla\gamma)$ the angle between $\mathbf{N}$ and $\nabla\gamma$. The effective diffusivity $K_f$ is conventionally defined using $K_i = 1000\,\text{m}^2.\text{s}^{-1}$, a notional isoneutral turbulent mixing coefficient typical of observed values. It has become conventional to use the value $K_f = 10^{-5}\,\text{m}^2.\text{s}^{-1}$ as the threshold separating acceptable from

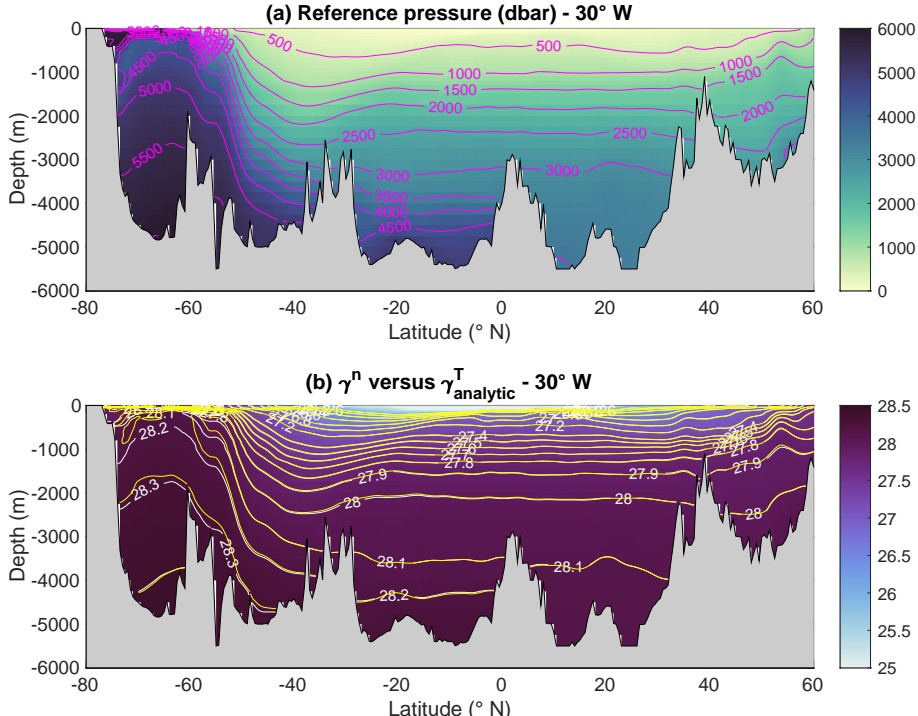

**Figure 3.** (a) Atlantic section along $30°W$ of the variable reference pressure $p_r = p_r(S_A,\Theta) = p_0(z_r)$ used for the construction of potential spiciness $\tau_\ddagger(S,\theta,p_r)$. (b) Atlantic section along $30°W$ of Jackett and McDougall (1997) empirical neutral density variable $\gamma^n$. The white labelled contours represent selected isolines of $\gamma^n$, while the unlabelled yellow contours represent the same isolines but for $\gamma^T_{analytic}$.

annoyingly large degree of non-neutrality. By this measure, Fig. 4 (b) shows that unlike $\sigma_1$ and $\sigma_2$, the degree of neutrality of $\gamma^T_{analytic}$ is uniformly acceptable everywhere in the water column north of $50°S$, where it is comparable to that of $\gamma^n$. How to adapt McDougall and Krzysik (2015) and Huang et al. (2018) potential spiciness and spicity variables to be consistent with $\gamma^T_{analytic}$ is discussed in the next section.

## 3   On the construction and estimation of potential spiciness-as-state-function variables

Since spiciness is a water mass property that can a priori be measured in terms of the isopycnal variations of any arbitrary function $\xi(S,\theta)$ independent of density, an important question in spiciness theory is whether there is any real physical justification or benefits for introducing the kind of dedicated spiciness-as-state-functions discussed by Veronis (1972), Jackett and McDougall (1985), Flament (2002), McDougall and Krzysik (2015), Huang (2011) and Huang et al. (2018). Assuming that this is the case, how can existing spiciness/spicity variables be adapted to be used consistently with $\gamma^T_{analytic}$ and the non-constant reference pressure $p_r(S,\theta)$ defined in the previous section, given that existing codes for computing such variables are in general limited to a few discrete reference pressures?

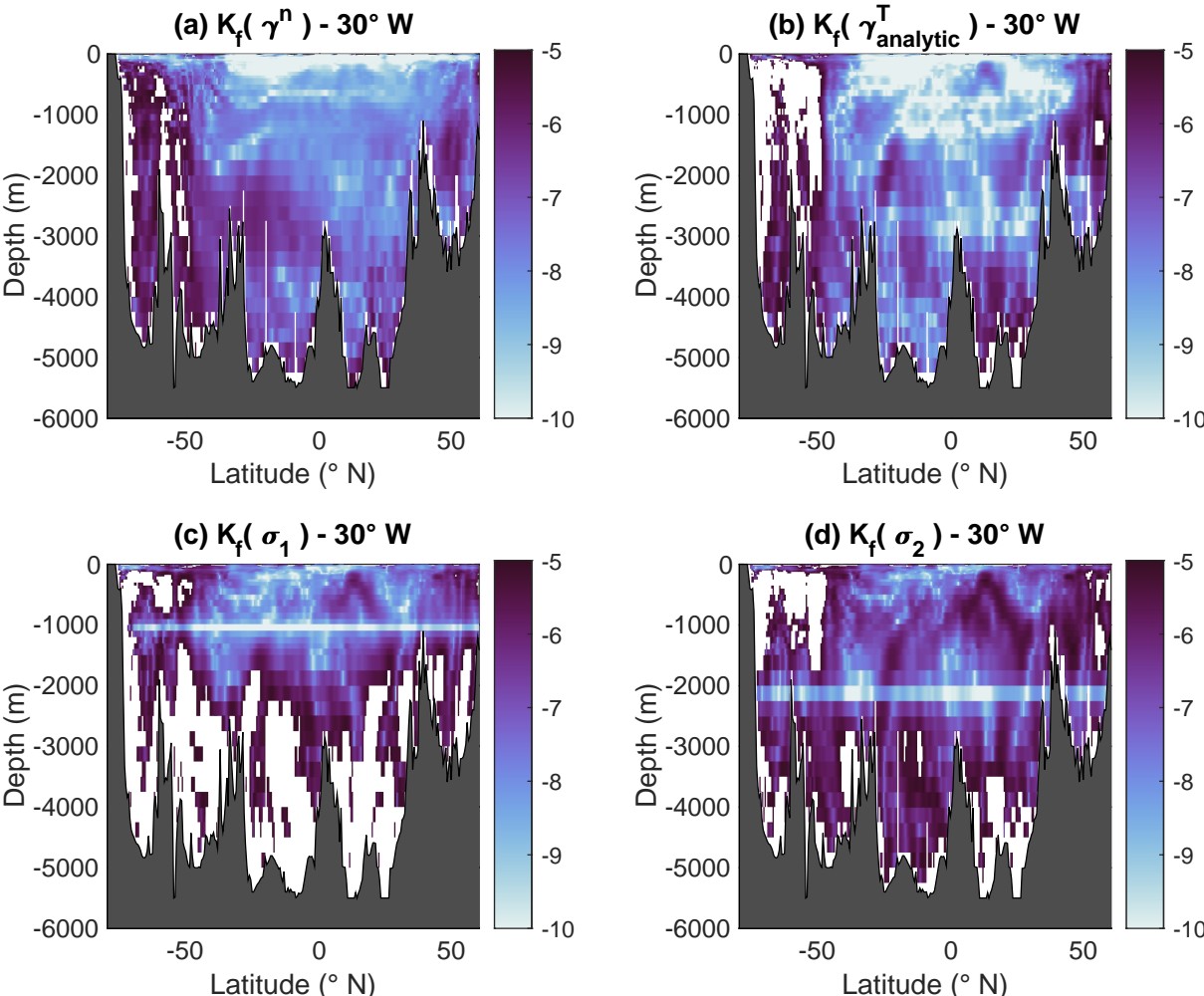

**Figure 4.** Atlantic sections along $30°W$ of the effective diffusivity of: (a) Jackett and McDougall (1997)'s empirical neutral density $\gamma^n$; (b) analytic thermodynamic neutral density $\gamma^T_{analytic}$; (c) potential density $\sigma_1$; (d) potential density $\sigma_2$. The white masked areas flag regions where $K_f > 10^{-5}\,\mathrm{m^2.s^{-1}}$, which is widely used as a threshold indicating large departure from non-neutrality. These show that $\gamma^T_{analytic}$ is in general significantly more neutral than standard potential density variables, while similarly neutral as $\gamma^n$ north of $50°S$.

## 3.1 Mathematical problems defining spiciness-as-state-functions

To examine the benefits that might be attached to a particular choice of spiciness-as-a-state function $\xi(S,\theta)$ for studying water mass contrasts, it is useful to establish some general properties about what controls its isopycnal variations. By denoting $d_i$ the restriction of the total differential operator to an isopycnal surface $\gamma = \mathrm{constant}$, the isopycnal variations of the latter may be

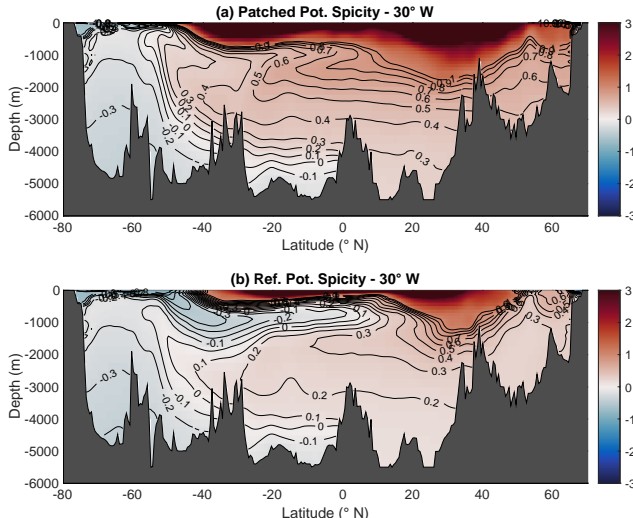

**Figure 5.** (a) Patched potential spicity referenced to the relevant reference pressure appropriate to the pressure range of fluid parcels; (b) potential spicity referenced to the variable reference pressure $p_r(S_A, \Theta)$. Referencing potential spicity to the variable reference pressure $p_r$ significantly increases its ability as a water mass indicator, especially with respect to the representation of AAIW.

written in the following equivalent forms:

$$\mathrm{d}_i\xi = \xi_S \mathrm{d}_i S + \xi_\theta \mathrm{d}_i \theta = \frac{J}{\gamma_S \gamma_\theta} \gamma_S \mathrm{d}_i S = -\frac{J}{\gamma_S \gamma_\theta} \gamma_\theta \mathrm{d}_i \theta$$

$$= \frac{J}{2\gamma_S \gamma_\theta} \left( \gamma_S \mathrm{d}_i S - \gamma_\theta \mathrm{d}_i \theta \right), \tag{10}$$

where $\xi_S = \partial \xi / \partial S$, $\xi_\theta = \partial \xi / \partial \theta$, $\gamma_\theta = \partial \gamma / \partial \theta$, $\gamma_S = \partial \gamma / \partial S$, using the fact that by construction $\gamma_S \mathrm{d}_i S + \gamma_\theta \mathrm{d}_i \theta = 0$, with $J = \partial(\xi, \gamma)/\partial(S, \theta)$ the Jacobian of the transformation $(S, \theta) \to (\xi, \gamma)$ as before. Eq. (10) establishes that:

1. $\mathrm{d}_i \xi$ is proportional to the elemental imperfect differential $\delta \tau = \gamma_S \mathrm{d}_i S = -\gamma_\theta \mathrm{d}_i \theta = \frac{1}{2}(\gamma_S \mathrm{d}_i S - \gamma_\theta \mathrm{d}_i \theta)$ with proportionality factor $J/(\gamma_S \gamma_\theta)$ for all spiciness-as-state-functions $\xi$. The two quantities $\gamma_S \mathrm{d}_i S$ and $\gamma_\theta \mathrm{d}_i \theta$ have the same physical units: they can thus be regarded as the basic building blocks for the construction of any spiciness-as-state-function variable if so desired;

2. $\mathrm{d}_i \xi$ is unaffected by the transformation $\xi \to \xi - \xi_r(\gamma)$, where $\xi_r(\gamma)$ is any arbitrary function of $\gamma$, so that both $\xi$ and $\xi - \xi_r(\gamma)$ have identical isopycnal variations. The benefit of imposing some particular property on $\xi$, such as orthogonality to $\gamma$, is therefore not obvious since such a property cannot in general be satisfied by both $\xi$ and $\xi - \xi_r(\gamma)$.

According to (10), the main quantity determining the properties of the spiciness-as-state-function $\xi$ is the Jacobian $J$. The generic mathematical problem determining $\xi$ may therefore be written in the form:

$$\gamma_\theta \frac{\partial \xi}{\partial S} - \gamma_S \frac{\partial \xi}{\partial \theta} = J(S, \theta). \tag{11}$$

Eq. (11) can be recognised as a standard quasi-linear partial differential equation amenable to the method of characteristics. Its general solution is defined up to some function $\xi_r(\gamma)$ depending on the boundary conditions imposed $\xi$. In the important particular cases where $\xi = \theta$ and $\xi = S$, the Jacobian and amplification factors are given by:

$$\xi = S \qquad \rightarrow \qquad J = \frac{\partial \gamma}{\partial \theta} \qquad \rightarrow \qquad \frac{J}{\gamma_S \gamma_\theta} = \frac{1}{\gamma_S}. \tag{12}$$

$$\xi = \theta \qquad \rightarrow \qquad J = -\frac{\partial \gamma}{\partial S} \qquad \rightarrow \qquad \frac{J}{\gamma_S \gamma_\theta} = -\frac{1}{\gamma_\theta}, \tag{13}$$

Eqs. (12) and (13) exemplify the two main kinds of behaviour of spiciness-as-state functions. For salinity-like $\xi$, the Jacobian varies as $\gamma_\theta$ and is therefore quite non-uniform, with loss of invertibility where $\gamma_\theta = 0$; however, the scaling factor $J/(\gamma_S \gamma_\theta)$ varies as $1/\gamma_S$ and therefore varies little in $(S,\theta)$ space. This is the opposite for temperature-like $\xi$, for which it is the Jacobian that is approximately constant and the scaling factor $J/(\gamma_S \gamma_\theta) = -1/\gamma_\theta$ that varies non-uniformly. To a large-extent, these opposing behaviours characterise McDougall and Krzysik (2015) and Huang et al. (2018) spiciness and spicity variables. Which behaviour is preferable cannot be determined without bringing in additional physical considerations discussed in Section 4 and conclusions.

## 3.2  Construction of reference potential spicity $\pi_{ref}$

Huang et al. (2018)'s spicity variable $\pi(S_A, \Theta)$ is designed to enforce Veronis (1972)'s definition of orthogonality in the re-scaled $S_A$ and $\Theta$ coordinates $X(S) = \rho_0 \alpha_0 \Theta$ and $Y(S_A) = \rho_0 \beta_0 S_A$, with $\alpha_0$ and $\beta_0$ representative constant values of the thermal expansion and haline contraction coefficients respectively. The imposition of this form of orthogonality ensures that the transformation $(S_A, \Theta) \rightarrow (\gamma, \pi)$ is invertible everywhere in $(S_A, \Theta)$ space, which is not the case of the $45°$ form of orthogonality considered by Jackett and McDougall (1985), Flament (2002) and McDougall and Krzysik (2015). Huang et al. (2018) provide a Matlab subroutine gsw_pspi(SA,CT,pr) to compute $\pi$ as function of Absolute Salinity and Conservative Temperature at the discrete set of reference pressure $p_r = 0, 500, 1000, 2000, 3000, 4000, 5000$ (in dbar). This provides suffi-cient vertical resolution for computing the reference potential spicity $\pi_{ref}(S_A, \Theta) = \pi(S_A, \Theta, p_r(S_A, \Theta))$ referenced to the variable reference pressure $p_r(S_A, \Theta)$ defined in the previous section using shape preserving spline interpolation, as illustrated in Fig. 5 (b). By comparison, panel (a) shows the patched potential spicity obtained by stacking up together potential spicity estimated at the reference pressure appropriate to its pressure range (which appears to be smoother than might have been ex-pected, given the six transition points of discontinuities at $p = 250, 750, 1500, 2500, 3500, 4500$ (dbar)). This shows that the use of the variable reference pressure $p_r(S_A, \Theta)$ has a dramatic impact on the ability of spicity to pick up the water mass signals of the Atlantic Ocean. This is especially evident for the AAIW signal, which is only vaguely apparent in patched potential spicity, while being nearly as well defined in $\pi_{ref}$ as in the salinity field. This suggests that it is more advantageous to use potential spicity with $\gamma_{analytic}^T$ than with patched potential density.

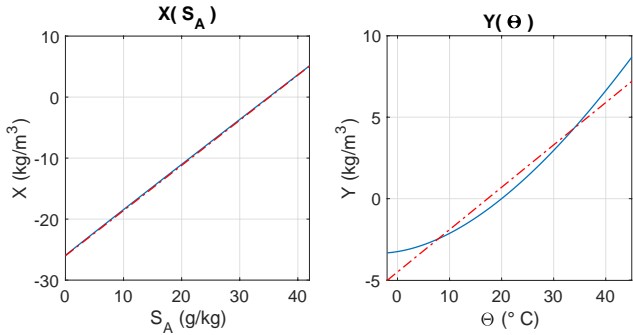

**Figure 6.** The re-scaled salinity and temperature $X(S_A)$ and $Y(\Theta)$ expressing both quantities in a common system of density-like units. Red dashed lines are the linear regressions of equation $X(S_A) = 0.74\,S_A - 26$ and $Y(\Theta) = 0.26\,\Theta - 4.5$.

### 3.3 Construction of reference potential spiciness $\tau_{ref}$

Jackett and McDougall (1985)'s spiciness-as-a-state function $\tau$ is designed so that its isopycnal variations are constrained to satisfy the following mathematically equivalent relations:

$$d_i\tau = 2\beta d_i S = 2\alpha d_i\theta = \beta d_i S + \alpha d_i\theta, \tag{14}$$

where, as before, $d_i$ is the restriction of the total differential operator to the isopycnal surface $\gamma(S,\theta) = \text{constant}$, where $\alpha$ and $\beta$ are the haline contraction and thermohaline expansion coefficients. In Jackett and McDougall (1985), density surfaces are approximated in terms of $\sigma_0$ but the use of patched potential density is implicit in the more recent paper by McDougall and Krzysik (2015). By comparing (14) with (10), it can be seen that Jackett and McDougall (1985) construction implies a constant proportionality factor $J/(\rho_S\rho_\theta) = 2$. This implies in turn for the Jacobian and quasi-linear PDE satisfied by $\tau$

$$J = \frac{2\rho_S\rho_\theta}{\rho} = -2\rho\alpha\beta \qquad \rightarrow \qquad \frac{\tau_S}{\beta} - \frac{\tau_\theta}{\alpha} = 2. \tag{15}$$

To solve the quasi-linear PDE (15), Flament (2002) and Jackett and McDougall (1985) further imposed $\tau$ to satisfy the so-called 45 degrees orthogonality, which in practice amounts to assume that the total differential of $\tau$ satisfy $d\tau \approx \lambda(\beta dS + \alpha d\theta)$, with $\lambda$ some integrating factor. Flament (2002) and Jackett and McDougall (1985) approached the problem somewhat differently, but their variables are nevertheless approximately linearly re-scaled functions of each other.

Both Jackett and McDougall (1985) and Flament (2002) provide polynomial expressions in powers of $S$ and $\theta$ for their potential spiciness variable, which are however limited to a single reference pressure $p_r = 0$. More recently, McDougall and Krzysik (2015) have provided Matlab subroutines (available at www.teos-10.org) `gsw_spice0(SA,CT)`, `gsw_spice1(SA,CT)` and `gsw_spice2(SA,CT)` to compute potential spiciness at the three reference pressures $p_r = 0$, $p_r = 1000\,\text{dbar}$ and $p_r = 2000\,\text{dbar}$, where SA is Absolute Salinity and CT Conservative Temperature. Because this limited number of reference pressures is far from ideal for computing potential spiciness $\tau(S_A,\Theta,p_r(S_A,\Theta))$ referenced to the variable reference pressure $p_r(S_A,\Theta)$ underlying $\gamma_{analytic}^T$, I have constructed an analytical proxy for McDougall and Krzysik (2015)'s variable valid for

the full range of pressures encountered in the ocean. The expression for this proxy, called quasi-linear spiciness, is

$$
\begin{aligned}
\tau_{\ddagger}(S,\theta,p) =& X(S,p) + Y(\theta,p) + \tau_0(p) \\
=& \rho_{00} \ln \left\{ \frac{\rho(S,\theta_0,p)}{\rho(S_0,\theta,p)} \right\} + \tau_0(p).
\end{aligned}
\tag{16}
$$

$\tau_{\ddagger}$ is a linear combination of the nonlinear re-scaled salinity and temperature coordinates $X(S,p)$ and $Y(\theta,p)$ and of reference function $\tau_0(p)$, whose expressions are:

$$
\begin{aligned}
X =& X(S,p) = \rho_{00} \ln \left\{ \frac{\rho(S,\theta_0,p)}{\rho(S_0,\theta_0,p)} \right\}, \\
Y =& Y(\theta,p) = -\rho_{00} \ln \left\{ \frac{\rho(S_0,\theta,p)}{\rho(S_0,\theta_0,p)} \right\},
\end{aligned}
\tag{17}
$$

$$
\tau_0(p) = -\rho_{00} \ln \left\{ \frac{\rho(S_{ref},\theta_0,p)}{\rho(S_0,\theta_{ref},p)} \right\}.
\tag{18}
$$

These functions depend on some arbitrary reference constants, specified as follows: $\rho_{00} = 1000\,\text{kg.m}^{-3}$ is chosen to give $\tau_{\ddagger}$ the same unit as density, while $\tau_0(p) = \tau(S_0,\theta_0,p)$ specifies the reference value of $\tau_{\ddagger}$ at the reference point $(S_0,\theta_0)$. In principle, $S_0$ and $\theta_0$ could also be made to depend on pressure $p$, but this complication is avoided for simplicity. Fig. 6 illustrates a particular construction of $X(S_A,p)$ and $Y(\Theta,p)$, based on the values $S_0 = 35\,\text{g/kg}$, $\Theta_0 = 20°C$ and $p = 0\,\text{dbar}$. The reference point $(S_{ref},\theta_{ref})$ defines where $\tau_{\ddagger}$ vanishes. The values $S_{ref} = 35.16504\,\text{g/kg}$ and $\Theta_{ref} = 0°C$ were used to fix the zero of $\tau_{\ddagger}$ as in McDougall and Krzysik (2015). Fig. 6 shows that while $Y(S_A)$ varies approximately linearly with $S_A$, $Y(\Theta)$ clearly varies nonlinearly with $\Theta$. Linear regression lines are also indicated to illustrate the departure from nonlinearity.

The total and isopycnal differentials of potential spicines $\tau_{\ddagger}(S,\theta,p_r)$ referenced to the constant reference pressure $p_r$ are:

$$
d\tau_{\ddagger} = \rho_{00}(\beta_0\,dS + \alpha_0\,d\theta), \qquad d_i\tau_{\ddagger} = \rho_{00}\left(\frac{\beta_0}{\beta} + \frac{\alpha_0}{\alpha}\right)\beta d_i S.
\tag{19}
$$

where $\beta_0 = \beta(S,\theta_0,p_r)$ and $\alpha_0 = \alpha(S_0,\theta,p_r)$. For $(S,\theta)$ close enough to the reference point $(S_0,\theta_0)$, $d_i\tau_{\ddagger} \approx 2\rho_{00}\beta d_i S$, which is equivalent to the differential problem that Jackett and McDougall (1985) set out to solve. An important difference with standard spiciness variables, however, is that the Jacobian associated with $\tau_{\ddagger}$ is given by

$$
J = -\rho_{00}\rho(\alpha\beta_0 + \alpha_0\beta) < 0,
\tag{20}
$$

and differs from zero everywhere in $(S,\theta)$ space. As to the pressure dependence of $\tau_{\ddagger}$, it is given by

$$
\begin{aligned}
\frac{\partial \tau_{\ddagger}}{\partial p} =& \rho_{00}\left(\kappa(S,\theta_0,p) - \kappa(S_0,\theta,p)\right. \\
& \left. - \kappa(S_{ref},\theta_0,p) + \kappa(S_0,\theta_{ref},p)\right),
\end{aligned}
\tag{21}
$$

where $\kappa(S,\theta,p) = \rho^{-1}\partial\rho/\partial p(S,\theta,p)$. Eq. (21) shows that $\partial_p\tau_{\ddagger}$ vanishes at the two reference points $(S_0,\theta_0)$ and $(S_{ref},\theta_{ref})$; it follows that by design, $\tau_{\ddagger}$ is only weakly dependent on pressure, and hence naturally quasi-material (that is, approximately conserved following fluid parcels in the absence of diffusive sources/sinks of $S$ and $\theta$).

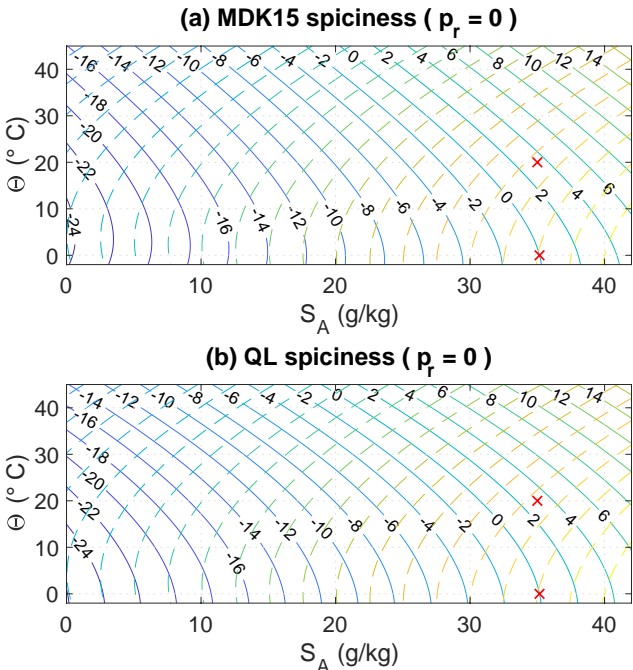

**Figure 7.** (a) Isocontours of McDougall and Krzysik (2015)'s spiciness variable referenced to the surface pressure (solid lines) along with $\sigma_0$ isocontours (dashed lines). (b) Isocontours of the mathematically explicit quasi-linear spiciness variable introduced in this paper referenced to the surface pressure (solid lines) along with $\sigma_0$ isocontours (dashed lines). The crosses indicate the reference point $(S_A, \Theta) = (35, 20)$ at which $X = Y = 0$, as well as the reference point $(S_A, \Theta) = (35.16504, 0.)$ at which both spiciness variables are imposed to vanish.

McDougall and Krzysik (2015)'s spiciness and $\tau_\ddagger$ are compared in Figs 7 and 8 in $(S_A, \Theta)$ space as well as in the re-scaled
$(X, Y)$ coordinates respectively at the reference pressure $p_r = 0$. The two variables can be seen to behave in essentially the same way, the result also holding at $p_r = 1000\,\mathrm{dbar}$ and $p_r = 2000\,\mathrm{dbar}$, except for cold temperature and low salinity values where the Jacobian associated with McDougall and Krzysik (2015) spiciness vanishes while that associated with $\tau_\ddagger$ does not. That both variables approximately satisfy the 45 degrees orthogonality is made obvious in the re-scaled $(X, Y)$ coordinates of Fig. 8; interestingly, this plot suggests that in-situ density is approximately a linear function of $X$ and $Y$, which is further examined
in Appendix B. Moving to physical space, Fig. 9 compares the patched potential spiciness computed using McDougall and Krzysik (2015) software (top panel) versus the reference potential spiciness $\tau_{ref} = \tau_\ddagger(S_A, \Theta, p_r(S_A, \Theta))$ calculated with $\tau_\ddagger$. For patched potential spiciness, the transition points of discontinuity were chosen at $p_r = 1000, 2000$ (dbar). As for potential spicity, it also appears to be advantageous to use potential spiciness with a variable $p_r$, as it makes AAIW more marked and NADW somewhat more homogeneous. The effect of a variable $p_r$ on potential spiciness is not as dramatic as for potential
spicity, however, which is likely due to the weak pressure dependence of $\tau_\ddagger$ noted above. In any case, the above analysis suggests that $\tau_\ddagger$ is a useful proxy for McDougall and Krzysik (2015)'s spiciness variable.

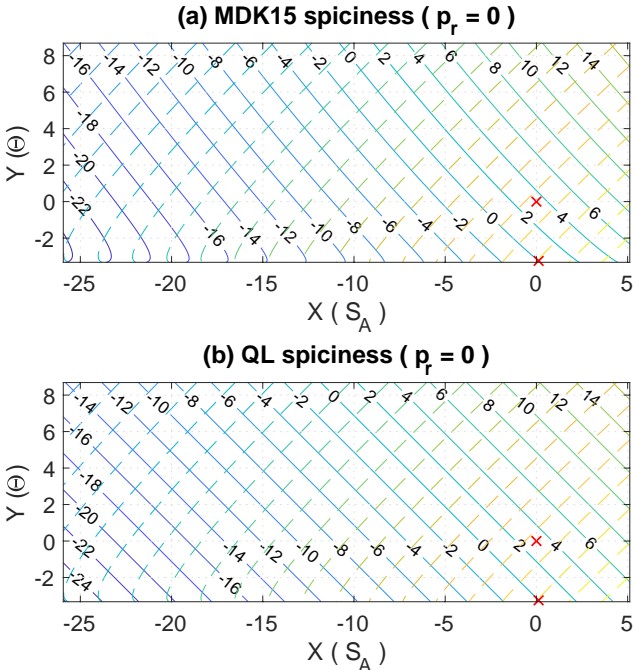

**Figure 8.** Same as Fig. 7 but with $S_A$ and $\Theta$ replaced by the re-scaled salinity and temperature coordinates $X(S_A)$ and $Y(\Theta)$, in which the spiciness and density variables visually appear approximately orthogonal to each other.

## 4  Links between spiciness-as-a-property, the zero of spiciness and 'orthogonality' in physical space

As stated previously, it is important to recognise that spiciness is not a substance but a property that cannot be described without taking into account empirical information about the particular water masses to be analysed. Whether this point is well known is
unclear, because the distinction between a property and a substance is rarely evoked if ever in the spiciness literature. As a result, it follows that the usefulness of spiciness-as-a-state-function variables such as those of Jackett and McDougall (1985), Flament (2002) or Huang et al. (2018), which do not incorporate any information about the particular water masses to be analysed, is only limited to quantifying the relative differences in spiciness for fluid parcels that belong to the same density surface, as pointed out Timmermans and Jayne (2016). In other words, any difference in spiciness for fluid parcels belonging to different
density surfaces predicted by such variables is physically meaningless. Physically, this limitation is associated with another key one, namely the impossibility to link a spiceless ocean to the zero value of a spiciness-as-a-state function variable. This is not surprising, because the zero and other isovalues of any spiciness-as-a-state-function are necessarily artificial since not informed by physical consideration about how to endow the relative spiciness of two fluid parcels that belong to different density surfaces with physical meaning. Here, a spiceless ocean is defined as a notional ocean in which iso-surfaces of potential temperature,
salinity and potential density would all coincide. It follows that in a spiceless ocean, any spiciness-as-a-state-function would have to be a function of density only, $\xi = \xi_r(\gamma)$ say, which in general must differ from zero. This suggests that spiciness-

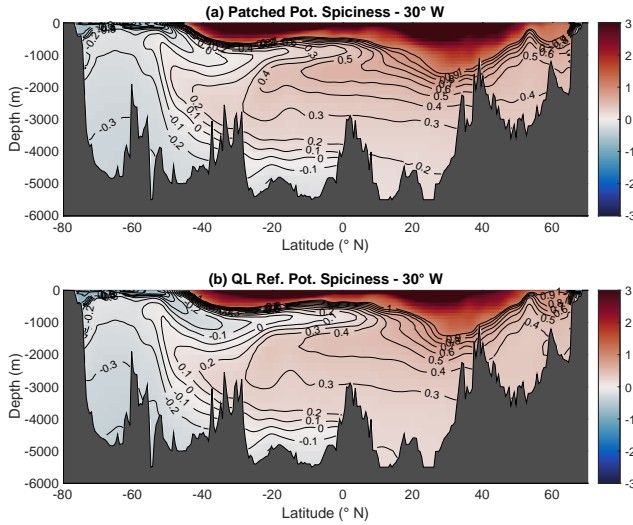

**Figure 9.** (a) Patched potential spiciness referenced to the 3 discrete reference pressures provided by McDougall and Krzysik (2015) software appropriate to their pressure ranges; (b) Quasi-linear potential spiciness referenced to the variable reference pressure $p_r = p_r(S_A, \Theta) = p_0(z_r)$. In contrast to spicity, the use of a variable reference pressure has little impact on the visual aspect of spiciness, thus confirming the weak pressure dependence of $\tau_\ddagger(S_A, \Theta, p)$.

as-a-property should be defined as the anomaly $\xi' = \xi - \xi_r(\gamma)$, as originally proposed by Jackett and McDougall (1985) and McDougall and Giles (1987). Clearly, such an approach addresses the zero-of-spiciness issue, since it ensures that $\xi'$ would vanish in a spiceless ocean, as desired. Moreover, it also define spiciness as a property rather than as a substance, because

although $\xi' = \xi'(S, \theta)$ is a function of $S$ and $\theta$, it is not truly a function of state owing to its dependence on the empirically determined function of density $\xi_r(\gamma)$.

Because any function $\xi(S, \theta)$ independent of $\gamma$ is a potential candidate for constructing a spiciness-as-a-property $\xi' = \xi - \xi_r(\gamma)$ variable, the following questions arise:

1. Are there any benefits in constructing a dedicated spiciness-as-a-state function $\xi$ for the purpose of constructing the

anomaly $\xi' = \xi - \xi_r(\gamma)$? Are there any good reasons to think that $S' = S - S_r(\gamma)$ or $\theta' = \theta - \theta_r(\gamma)$ are not suitable or insufficient for all practical purposes?

2. Since neither Veronis (1972)'s form of orthogonality nor Jackett and McDougall (1985)'s 45 degrees orthogonality can be satisfied by $\xi'$, what is the physical justification for imposing it on $\xi$ in the first place?

3. In physics, the scale to measure some quantities is commonly accepted to be arbitrary, which is why several scales (e.g.,

Kelvin, Celsius and Farenheit) have been developed over time to measure temperature for instance. To what extent is the problem of quantifying spiciness-as-a-property similar or different from that of constructing a temperature scale?

4. To what extent is the problem of deciding in favour of a particular choice of $\xi$ one that can be constrained by physical arguments as opposed to one that is fundamentally arbitrary and therefore a matter of personal preference?

5. Are there any additional constraints that $\xi' = \xi - \xi_r(\gamma)$ should satisfy beyond the zero-of-spiciness issue in order for relative difference in $\xi'$ for fluid parcels that belong to different density surfaces to be considered physically meaningful?

Jackett and McDougall (1985) shed some light on some of the above issues by suggesting that the choice of $\xi$ may be less important than one would think. Indeed, they showed that the inter-differences in appropriately re-scaled anomaly functions $\tau'_{jmd}$, $\tau'_\nu$ and $S'$ were considerably reduced over those exhibited by $\tau_{jmd}$, $\tau_\nu$ and $S$, a potentially important result that does not appear to have received much attention so far. Jackett and McDougall (1985)'s conclusion is only based on the comparison of two vertical soundings, however, so it is necessary to examine it more systematically in order to assess its robustness. To that end, I constructed particular examples of anomaly functions $\xi'$ for the 4 variables considered in the introduction, based on the use of second order polynomial descriptor for $\xi_r(\gamma)$ obtained by means of a nonlinear regression of $\xi$ against $\gamma^T_{analytic}$. A polynomial descriptor was preferred over using a simple isopycnal average as in Jackett and McDougall (1985) or McDougall and Giles (1987) to ensure the smoothness and differentiability of $\xi_r(\gamma^T_{analytic})$. To minimise the impact of outliers, the robust bisquare least-squares provided by Matlab was used. [3]. Because a scatter plot of $\xi$ against $\gamma^T_{analytic}$ does not reveal any particular relation between the two variables if all the ocean data points are used, as shown in Fig. 10 by the yellow points, the nonlinear regression for obtaining the second order polynomial $\xi_r(\gamma^T_{analytic})$ was restricted to the points making up the $30°W$ Atlantic section (indicated in red/orange), for which a relation is more apparent. The resulting function, depicted as the black solid line in Fig. 10, was then used to construct $\xi'$ both in thermohaline space in Fig. 11 and in physical space in Fig. 12. Similarly as in Jackett and McDougall (1985), Fig. 11 shows that the significant inter-differences in behaviour exhibited by the different $\xi$ are considerably reduced for $\xi'$. The result is significantly more general than in Jackett and McDougall (1985), however, since it pertains to a large part of $(S_A, \Theta)$ space as opposed to being limited to a few vertical soundings. In all plots of Fig. 11, $\xi'(S_A, \Theta)$ is shown to be an increasing function of $S_A$ but decreasing function of $\Theta$, similarly as density but with a much weaker dependence on $\Theta$. This departs from the conventional wisdom that spiciness should be an increasing function of both $S_A$ and $\Theta$, as is the case of McDougall and Krzysik (2015) spiciness and Huang et al. (2018) spicity variables. However, this is based on only one particular construction of $\xi_r(\gamma)$ constrained by the properties of the Atlantic Ocean, so might not be a general result valid for all possible constructions of $\xi_r(\gamma)$.

To make them more comparable, the various $\xi'$ were re-scaled by the r.m.s.e of all data points making up the $30°W$ Atlantic section before being drawn in Fig. 12. As a result of the re-scaling, the inter-differences between the different variables have been significantly reduced but some are still noticeable. For instance, panel (d) suggests that part of AAIW is entrained by the sinking of AABW, which is not really apparent in the other panels. All re-scaled variables now appear to perform equally well as water mass indicators, with all four main Atlantic water masses following similar patterns and being characterised by similar spiciness values in all plots. Thus, MIW appears as a high-spiciness water mass with values greater than about 0.3;

---

[3]The Matlab documentation page reviewing the various least-squares methods can be consulted at:

https://uk.mathworks.com/help/curvefit/least-squares-fitting.html

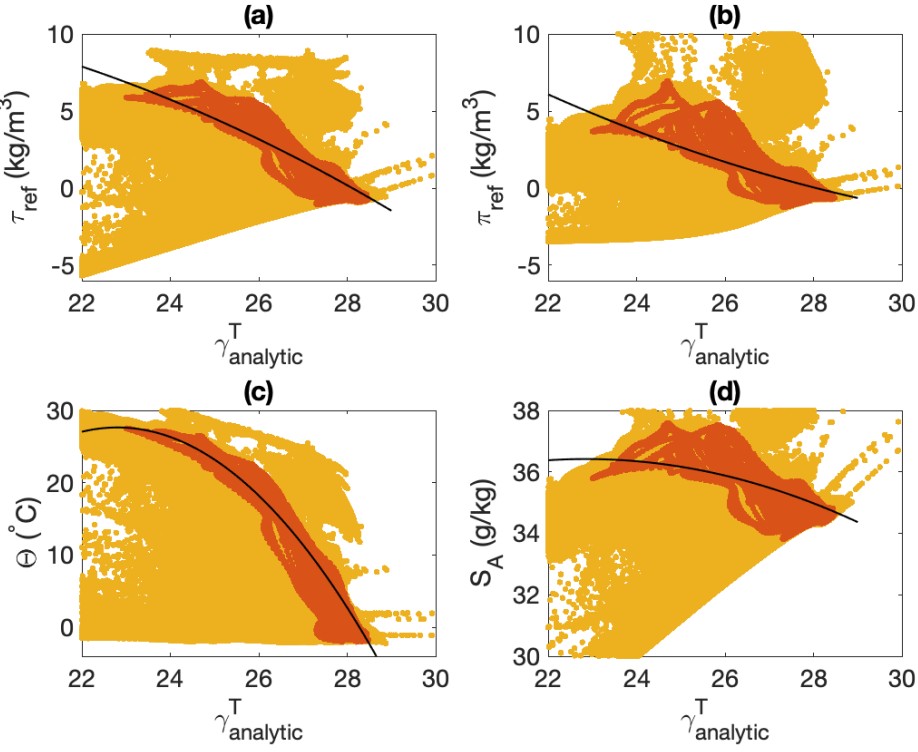

**Figure 10.** Nonlinear regression between $\gamma_{analytic}^T$ and various spiciness-as-state-functions estimated for data restricted to the $30^\circ W$ Atlantic section (in red/orange): (a) Potential reference spiciness $\tau_{ref}$; (b) Potential reference spiciness $\pi_{eref}$; (c) Conservative Temperature; (d) Absolute Salinity. The nonlinear regression curve is indicated in solid black and is described by a second order polynomial in $\gamma_{analytic}^T$. The yellow data points represent the whole dataset for the global ocean for comparison.

AAIW appears as a very low spiciness water mass with negative values; AABW and NADW appear as water masses differing only by about 0.1 spiciness units, with values for AABW and NADW ranging from 0 to about 0.2 for the former, and from 0.2 to about 0.3 for the latter, except for $\Theta'$ for which it goes to about 0.2. The nature of the inter-differences between the various re-scaled $\xi'$ is further clarified by plotting each variable against each other as shown in Fig. 13. Panel (c) reveals that $\tau'_{ref}/\Delta\tau$ and $\Delta S'_A/\Delta S$ are essentially indistinguishable from each other; however, the other panels reveal that $\Theta'/\Delta\Theta$ and $\pi'_{ref}/\Delta\pi$ are nonlinear multi-valued functions of $S'_A/\Delta S_A$ (or $\tau'_{ref}/\Delta\tau$). It follows that the strong similarities exhibited by all re-scaled anomaly variables in Fig. 12 hide some important fundamental differences.

In the introduction, I showed that the superior ability of salinity to pick up the different water mass signals of the Atlantic ocean [4] appeared to coincide with being the most 'orthogonal' to $\gamma_{analytic}^T$, orthogonality being defined in terms of the median of all the angles between $\nabla\xi$ and $\nabla\gamma_{analytic}^T$ estimated for all available data points. As shown in Fig. 2, such a metric ranks the

---

[4]as based on an arguably subjective visual comparison of the different panels of Fig. 1

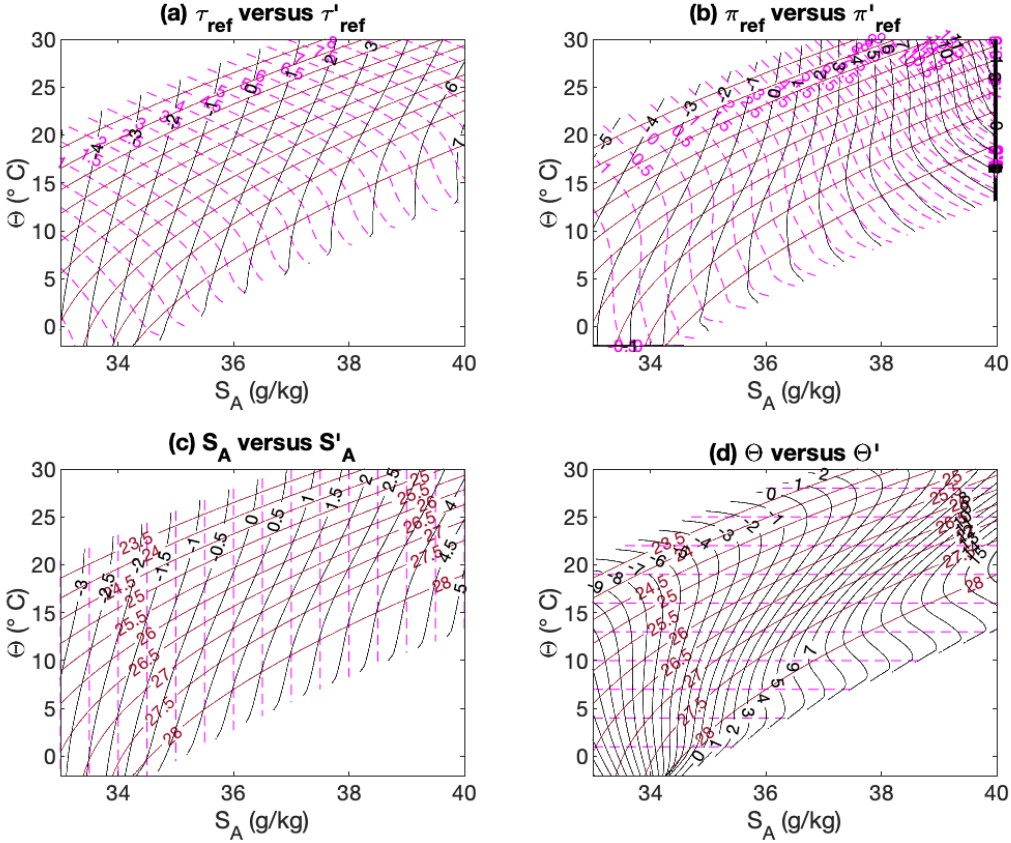

**Figure 11.** Isocontours of $\gamma^T_{analytic}$ (red/brown solid lines), $\xi$ (dashed magenta lines), and $\xi'$ (black solid lines) for various spiciness-as-state functions $\xi$: (a) Reference potential spiciness $\tau_{ref}$; (b) Reference potential spicity $\pi_{ref}$; (c) Absolute Salinity; (d) Conservative Temperature. The isocontours for $\gamma^T_{analytic}$ are the same in all panels but labels are only shown in panels (c) and (d).

various $\xi$ in the same order as a visual determination of their ability as water mass indicators. Since it seems clear from Fig. 12 that $\xi'$ systematically improves over $\xi$ as a water mass indicator, one may ask whether this improvement can be similarly attributed to $\xi'$ being more orthogonal to $\gamma^T_{analytic}$ than $\xi$. Fig. 14 illustrates a particular idealised example for which this would be expected, as in this case $\nabla\xi'$ can actually be imposed to be exactly orthogonal to $\nabla\gamma$, viz.,

$$\nabla\gamma \cdot \nabla(\xi - \xi_r(\gamma)) = 0, \tag{22}$$

provided that $\xi_r(\gamma)$ is chosen to satisfy:

$$\xi'_r(\gamma) \approx \frac{\nabla\xi \cdot \nabla\gamma}{|\nabla\gamma|^2}. \tag{23}$$

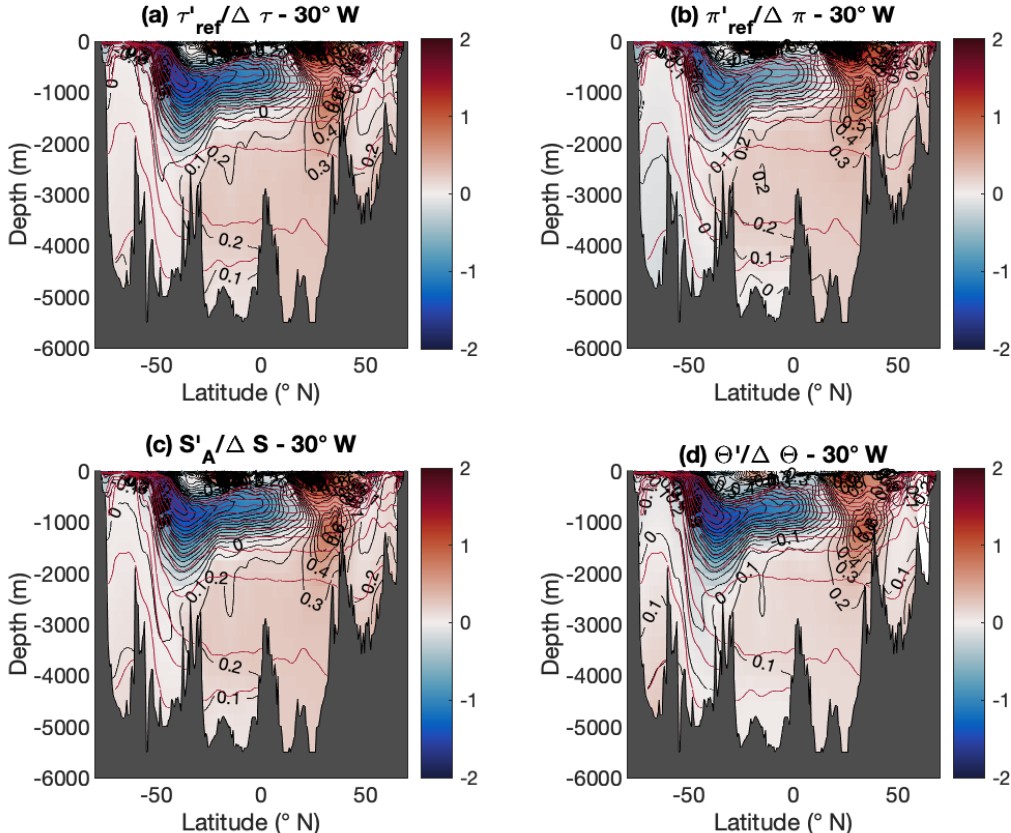

**Figure 12.** Atlantic Ocean sections along $30°W$ of the normalised spiciness anomaly functions $\xi'/\Delta\xi = (\xi - \xi_r(\gamma^T_{analytic}))/\Delta\xi$, with $\xi_r(\gamma^T_{analytic})$ corresponding to the nonlinear regression functions depicted in Fig. 10, for: (a) Reference potential spiciness $\tau_{ref}$; (b) Reference potential spicity $\pi_{ref}$; (c) Absolute Salinity; (d) Conservative Temperature. Brown solid lines represent the same selected isopycnal contours for $\gamma^T_{analytic}$ as in Fig. 1. Each variable has been re-scaled by the r.m.s.e. $\Delta\xi$ of all data points making up this section before being drawn.

However, Fig. 2 reveals that although the orthogonality of $\xi'$ to $\gamma^T_{analytic}$ is occasionally improved over that of $\xi$, this is not systematically the case, which suggests that the idealised case of Fig. 14 is not representative of the relative distributions of $\xi$ and $\gamma^T_{analytic}$ for the actual ocean water masses. In fact, the values of the median angles illustrated in Fig. 2 rarely exceed $0.02$ degrees, which means that the isolines of $\xi$ or $\xi'$ tend more often than not to make a small angle with isopycnal surfaces and that $\nabla\xi'$ is only truly orthogonal to $\nabla\gamma^T_{analytic}$ in a few frontal regions where two water masses collide. This is the case for instance where AAIW meets MIW around $20°N$, as can be seen in Fig. 12. This suggests, therefore, that the angle between $\nabla\xi'$ and $\nabla\gamma$ is strongly controlled by isopycnal stirring outside such frontal regions. Therefore, although the orthogonality in

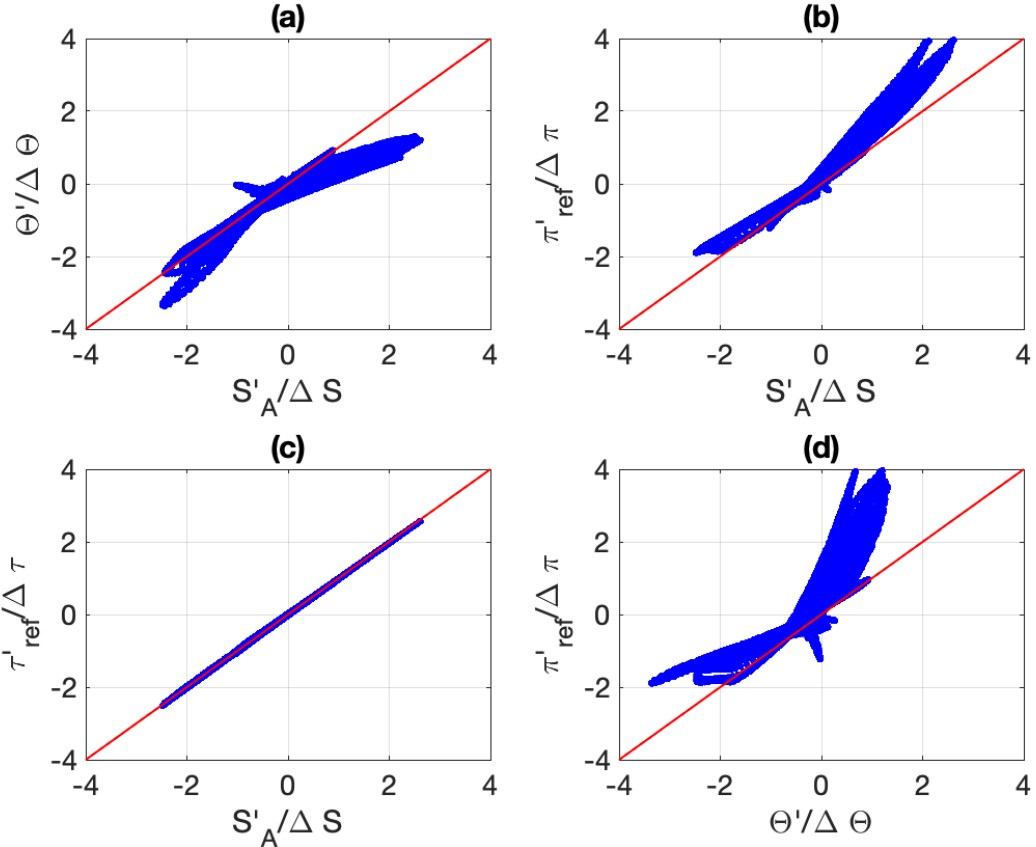

**Figure 13.** Scatter plots of various normalised $\xi'$ against each other. Panel (d) shows that $\tau'_{ref}/\Delta\tau$ is essentially equivalent to $S'_A/\Delta S$. The other panels suggest that $\pi'_{ref}$ and $\Theta'$ behave in fundamentally different ways than $S'_A$ and $\tau'_{ref}$ for positive and negative spiciness values.

physical space seems more germane to spiciness theory than orthogonality in thermohaline space, how to actually make use of it to constrain $\xi$ or $\xi_r(\gamma)$ has yet to be fully understood and clarified.

## 5 Conclusions

In this paper, I have revisited the theory of spiciness and established that its main ingredients are: 1) a quasi-material density-like variable $\gamma(S,\theta)$ that needs to be as neutral as feasible; 2) a quasi-material spiciness-as-a-state function $\xi(S,\theta)$ independent of $\gamma$, so that $(\xi,\gamma)$ can be inverted to recover the $(S,\theta)$ properties of any fluid parcel; 3) an empirical reference function $\xi_r(\gamma)$ defined such that the zero value of the spiciness-as-a-property $\xi' = \xi - \xi_r(\gamma)$ describes a physically plausible notional spiceless

ocean in which all surfaces of constant salinity, potential temperature and density would coincide.

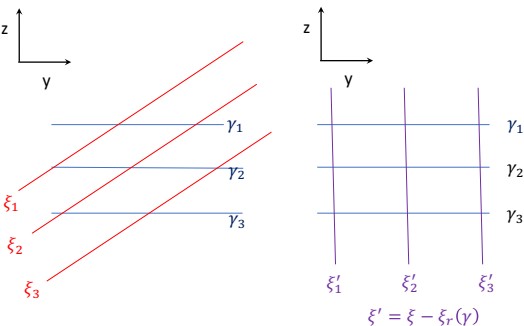

**Figure 14.** Schematics of the effect of subtracting a suitably defined function of density $\gamma$ from a spiciness-as-state-function $\xi$. In the left panel, the isolines of $\gamma$ and $\xi$ are assumed to be linear functions of $z$, such that $\gamma = az + b$ and $\xi = cz + dy$ and at an angle less than $90°$. In the right panel, the isolines of $\xi' = \xi - \xi_r(\gamma)$ are described by the equation $\xi' = dy = \text{constant}$, with $\xi_r(\gamma) = c[\gamma - b]/a$. This has removed the $z$-dependent part of $\xi$, resulting in $\xi'$ and $\gamma$ to be orthogonal in physical space.

Ingredient 1) is required because contrary to what has been assumed by some authors, it is not the properties of $\xi$ that determines its degree of dynamical inertness but the degree of neutrality of $\gamma$, regardless of what $\xi$ is. This result is important because it establishes that the theory of spiciness is not independent of the theory of isopycnal analysis. In particular, it provides a rigorous theoretical justification for the pursuit of a globally defined material density-like variable maximising neutrality as originally proposed by Eden and Willebrand (1999) as an alternative to Jackett and McDougall (1997) empirical neutral density variable $\gamma^n$. In this paper, I have proposed the use of a new implementation of Tailleux (2016b)'s thermodynamic neutral density $\gamma^T$, which is smoother, more neutral and computationally simpler to estimate than the original construction. As far as I am aware, this variable is currently the most neutral material density-like variable available. I also showed how to adapt McDougall and Krzysik (2015) and Huang et al. (2018) variables to construct the relevant forms of potential spiciness $\tau_{ref}$ and potential spicity $\pi_{ref}$ referenced to the variable reference pressure $p_r(S,\theta)$ underlying the construction of $\gamma^T$. Interestingly, it is found that the use of $p_r$ improves both potential spicity and potential spiciness as water mass indicators, although this improvement is more evident for the former than for the latter.

One of the main key points repeatedly emphasised in this paper is that spiciness is not a substance but a property that cannot be meaningfully defined independently of the particular ocean water masses to be analysed. This is why spiciness-as-a-property is really best measured by the anomaly $\xi' = \xi - \xi_r(\gamma)$ rather than by the spiciness-as-state-function $\xi$, where the empirical information about the water masses analysed is encoded in the reference function $\xi_r(\gamma)$. Although the use of spiciness anomalies was recommended early on by Jackett and McDougall (1985) and McDougall and Giles (1987), and underlies most of the literature devoted to understanding the role of spiciness on climate, it has since become completely overlooked in the most recent literature about spiciness theory in favour of the mathematical aspects pertaining to the construction of spiciness-as-state-functions orthogonal to density, e.g., Flament (2002), McDougall and Krzysik (2015), Huang (2011), Huang et al. (2018). This paper argues that the development of any dedicated spiciness-as-a-state function $\xi$ only makes sense if it is accompanied

by a description of the reference function $\xi_r(\gamma)$ that is needed to construct the anomaly $\xi' = \xi - \xi_r(\gamma)$. However, the fact that the normalised form of $\xi'$ does not appear to be very sensitive to the choice of $\xi$, as first suggested by Jackett and McDougall (1985) and illustrated by Figs. 11 and 12, combined with the fact than any form of orthogonality in $(S, \theta)$ space that may have been imposed on $\xi$ is lost by $\xi'$, raise questions about the actual need of dedicated spiciness-as-state-functions for constructing suitable water masses indicators or for studying the role of spiciness on climate.

One key advantage of $\xi'$ over $\xi$, however $\xi$ is defined, is that a notional spiceless ocean can be associated to the zero value of $\xi'$, which in turn allows one to give physical meaning to differences in $\xi'$ for fluid parcels belonging to different density surfaces, neither of which make sense for $\xi$. This view is supported by Fig. 12, which shows that all the Atlantic water masses appear to have the same kind of spiciness behaviour regardless of $\xi'$, namely: AAIW has very low spiciness, MIW has very high spiciness, while AABW and NADW have intermediate spiciness. Although differences in the way each variable represent ocean water masses exist, they tend to be rather subtle, making it hard to decide whether one variable should be regarded as superior to the others. Jackett and McDougall (1985) appear to disagree, as they have argued that the physical basis for their variable $\tau_{jmd}$ makes $\tau'_{jmd}$ superior to $S'$ and $\tau'_\nu$ for measuring water mass contrasts. However, they did not elaborate much on the physical meaning of this assumed superiority nor on how their claim could be independently tested. A clearer and physically more transparent way to assess the relative merits of a given $\xi'$ would be by establishing whether it mixes linearly or nonlinearly under the action of irreversible diffusive mixing (a variable mixes linearly if the spiciness of the mixture is equal to the mass weighted average of the individual spiciness values). If $S$ and $\theta$ are governed by a standard advection/diffusion equation with symmetric turbulent mixing tensor $\mathbf{K}$, the equation for $\xi'$ will be of the form:

$$\frac{D\xi'}{Dt} = \nabla \cdot (\mathbf{K}\nabla\xi') + \dot{\xi}'_{irr}, \tag{24}$$

$$\xi'_{irr} = \xi'_{SS}(\mathbf{K}\nabla S) \cdot \nabla S + \xi'_{\theta\theta}(\mathbf{K}\nabla\theta) \cdot \nabla\theta + 2\xi'_{S\theta}(\mathbf{K}\nabla S) \cdot \nabla\theta. \tag{25}$$

For $\xi'$ to mix linearly, the nonconservative production/destruction term $\xi'_{irr}$ due to its nonlinearities in $S$ and $\theta$ needs to be small. In this regard, $S'_A$ and $\tau_{ref}$ appear to be the variables in Fig. 11 that exhibit the smallest degree of nonlinearities in $S_A$ and $\Theta$, which suggests that they might be more conservative than $\pi_{ref}$ and $\Theta'$, although this remains to be checked. In this paper, $\xi_r(\gamma)$ was defined as a second order polynomial in $\gamma$ so as to guarantee smoothness and differentiability, which is not necessarily true of $\xi_r(\gamma)$ defined in terms of an isopycnal average for instance. A full discussion of whether alternative ways to construct $\xi_r(\gamma)$ might be preferable is beyond the scope of this paper. For instance, one could ask the question of whether it is possible to construct $\xi$ and $\xi_r(\gamma)$ so that $\xi'$ is as conservative as possible. Another important question is whether constraining $\xi$ to be orthogonal to $\gamma$ in thermohaline space, as pursued by McDougall and Krzysik (2015) or Huang et al. (2018), yields any special benefit for $\xi'$. To what extent can orthogonality in physical space be useful? Hopefully, the present work will help stimulate further research on these issues.

*Code availability.* Matlab and Python subroutines for computing the variable reference pressure $p_r(S_A, \Theta)$ and $\gamma_{analytic}^T$ used in this paper can be obtained by emailing the author.

**Table A1.** Coefficients for analytical reference density profile $\rho_0(z)$.

| parameters | value |
|---|---|
| a | 4.56016575 |
| b | -1.24898501 |
| c | 0.00439778209 |
| d | 1030.99373 |
| e | 8.32218903 |

**Table A2.** Coefficients for the polynomial function $f(p)$.

| coeff | value | confidence interval |
|---|---|---|
| a1 | 0.0007824 | (0.0007792, 0.0007855) |
| a2 | -0.008056 | (-0.008082, -0.008031) |
| a3 | 0.03216 | (0.03209, 0.03223) |
| a4 | -0.06387 | (-0.06393, -0.06381) |
| a5 | 0.06807 | (0.06799, 0.06816) |
| a6 | -0.03696 | (-0.03706, -0.03687) |
| a7 | -0.08414 | (-0.08419, -0.0841) |
| a8 | 6.677 | (6.677, 6.677) |
| a9 | 6.431 | (6.431, 6.431) |

**Appendix A: Definition and construction of $p_r(S,\theta)$ and $\gamma^T_{analytic}(S,\theta)$**

The values entering the definition of the reference density profile $\rho_0(z)$ are given in Table A1, whereas the coefficients for the polynomial entering the construction of $\gamma^T_{analytic}$ are given in Table A2.

**Appendix B: Quasi-linear approximation to in-situ density**

The re-scaled salinity/temperature coordinates given by Eq. (17) make it possible to construct a quasi-linear approximation
$\rho_{\ddagger} = \rho_{\ddagger}(S, \theta, p; S_0, \theta_0)$ of in-situ density as follows:

$$
\begin{aligned}
\rho_{\ddagger} &= \frac{\rho_0(p)}{\rho_{00}}(X - Y + \rho_{00}) \\
&= \rho_0(p)\left[\ln\left\{\frac{\rho(S,\theta_0,p)\rho(S_0,\theta,p)}{\rho_0^2(p)}\right\} + 1\right],
\end{aligned}
\tag{B1}
$$

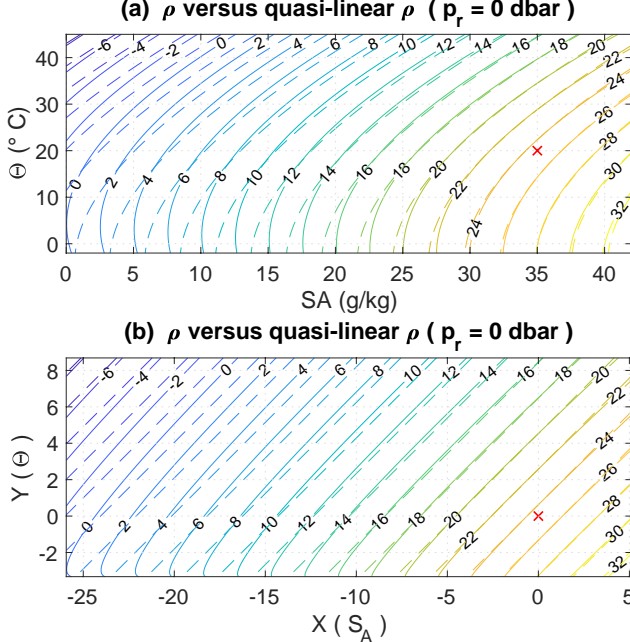

**Figure B1.** Comparison between potential density (referenced at $p = 0$ (dbar) (solid line) and its quasi-linear approximation (dashed line), seen as function of $S_A$ and $\Theta$ (top panel) and re-scaled coordinates $X$ and $Y$ (bottom panel). The red cross denotes the point $(S_A, \Theta) = (35., 20.)$ at which the two functions are imposed to be equal.

where $\rho_0(p) = \rho(S_0, \theta_0, p)$, so that by construction, $\rho_\ddagger = \rho$ at the reference point $(S_0, \theta_0)$ for all pressures. In-situ density and its quasi-linear approximation are compared in Fig. B1 for $p = 0$, as a function of $S_A$ and $\Theta$ (top panel) as well as of $X$ and $Y$ (bottom panel), with the red cross indicating the reference value $(S_A = 35\,\mathrm{g/kg}, \Theta = 20°C)$ used in the definition of $\rho_\ddagger$. As

expected, the accuracy of $\rho_\ddagger$ decreases away from the reference point, but appears to be reasonable in the restricted salinity range $[30\,\mathrm{g/kg}, 40\mathrm{g/kg}]$ that pertains to the bulk of ocean water masses. Interestingly, the bottom panel of Fig. B1 reveals that a significant fraction of the nonlinear character of the equation of state is captured by $X$ and $Y$, so that $\rho$ appears to be approximately linear in such coordinates.

The accuracy of the quasi-linear approximation $\rho_\ddagger$ can also be evaluated by examining how its thermal expansion, haline

contraction and compressibility compare with that of in-situ density. These are given by:

$$\alpha_\ddagger = -\frac{1}{\rho_\ddagger}\frac{\partial \rho_\ddagger}{\partial \theta} = \frac{\rho_0(p)\alpha(S_0, \theta, p)}{\rho_\ddagger}, \tag{B2}$$

$$\beta_\ddagger = \frac{1}{\rho_\ddagger}\frac{\partial \rho_\ddagger}{\partial S} = \frac{\rho_0(p)\beta(S, \theta_0, p)}{\rho_\ddagger}, \tag{B3}$$

$$\kappa_{\ddagger} = \frac{1}{\rho_{\ddagger}} \frac{\partial \rho_{\ddagger}}{\partial p} = \kappa(S_0, \theta_0, p)$$

$$+ \frac{\rho_0(p)}{\rho_{\ddagger}} \left[ \kappa(S, \theta_0, p) + \kappa(S_0, \theta, p) - 2\kappa(S_0, \theta_0, p) \right]. \tag{B4}$$

These relations show that the first partial derivatives of $\rho_{\ddagger}$ with respect to its three variables also coincide with their exact values at the reference point $(S_0, \theta_0)$, with the accuracy of the approximations decaying away from it, as expected.

*Author contributions.* The author performed all the work, analysis, and writing up.

*Competing interests.* The author declares no competing interests.

*Acknowledgements.* This work was supported by the NERC-funded OUTCROP project NE/R010536/1. The comments of Dr. Jan Zika, and two anonymous reviewers, as well as the technical editing of Prof. Ilker Fer, greatly helped improving clarity of the manuscript and are gratefully acknowledged.

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
