# Peer review of "Spiciness theory revisited, with new views on neutral density, orthogonality and passiveness"

_Ocean Science, 2020_

## Referee Comment (RC1) · Anonymous Referee #1 · 16 Jun 2020

This manuscript aims at clarifying the long-debated definition of a passive variable along neutral/isopycnal layers, commonly referred as "spiciness". The paper clarifies and demonstrates that the use of thermohaline anomalies (in particular absolute salinity) along neutral surfaces is sufficient to provide orthogonality in physical space. The long sought orthogonality in thermohaline space is showed to be flawed and not necessary to construct an inert variable along neutral surfaces. Moreover, the author discusses and resolves several issues raised by the definition of a physical variable satisfying the properties of spiciness. The existence of neutral surfaces is revealed to be key to the construction of a spiciness-like variable. By using theoretical arguments and a quasi-linear transformation of T/S space, the author also compares published definitions based on different assumptions and unifies them under basic principles.

I found the manuscript very interesting and well written. It surely provides an important step forward to the study of water mass. I therefore only have a few minor comments and recommend this paper to be published.

I60 and Fig 1 : What is the source of the data shown?

Fig2 : In caption :  $\sin(\nabla \sigma_1, \nabla \xi)$  ? Why is the yellow histogram closer to 0 (ie sine closer to 1, angle closer to  $\pi/2$ ), but described as the less orthogonal? Have the blue and yellow histograms been swapped? How does  $\Theta$  variable compare to  $S_A$  in terms of orthogonality?

1149 : What would be the proportion of the world ocean covered in that range? To identify regions where a spiciness definition would be challenged could be an interesting add to the paper.

- Along the manuscript, it is commonly referred to "orthogonality in physical space" and I think it would be nice to have a clear definition of what it means in introduction.

- I have the feeling that in regions of the ocean with temperature-driven density, salinity anomalies will have be a better choice to construct an inert variable. Am I speculating too much? Would  $\Theta'$  be any better than  $S'_A$  where density is salinity-driven (eg, coastal ocean, near sea-ice, Mediterranean, Red, Black Seas, ...)?

OSD

---

## Referee Comment (RC2) · Anonymous Referee #2 · 21 Jun 2020

**General Comments**

This manuscript aims to clarify the theoretical foundation for a spiciness variable sought by many oceanographers. The main idea is that, before considering spiciness, one must first construct a good neutral density variable that is materially conserved, and then most any materially conserved function can be used to construct a spiciness variable, simply by constructing its anomaly along neutral surfaces. The author also clarifies that pursuing orthogonality of spiciness and neutral density in $S-\Theta$ space is misguided, and instead that the goal should be orthogonality of their gradients in physical space.

Unfortunately, many of the advances of this paper are overstated, either lacking justifi-

cation, detail, or novelty. There are several logical errors as well. These are discussed below. I believe this manuscript has the potential to nicely tie together the theory of spiciness variables, but Major Revisions are required to get there.

One of the major points of the paper, that what matters for spiciness is actually the neutral density variable $\gamma$, was made by Jacket and McDougall (1985). The author has acknowledged this in some places, but a reader could easily get the impression that this idea owes to this manuscript. A stand-out example is in the abstract (line 5): stating "contrary to what is usually assumed" is unfair. Anyone who has read Jackett and McDougall (1985) would not assume this. This phrase should be removed, and a citation to Jackett and McDougall (1985) given in the abstract.

I find Fig 11 the most interesting aspect of this work. It is essentially a global test of the Jackett and McDougall (1985) idea, repeated here, that it is the anomaly $\xi'$ that is dynamically inert. The author's anomaly is defined as relative to a global isopycnal average. The results are evidently meaningful, but it is not entirely clear that more refined results could be obtained by refining the averaging procedure. McDougall and Giles (1987) argued in favor of studying property (salinity) anomalies relative to a *local* isopycnal average. To study a particular water mass intrusion, the state of the ocean far away should be irrelevant. It would therefore be prudent of the author to discuss the utility of using global isopycnal averages, and to locate the present work relative to the earlier work of McDougall and Giles (1987).

Moreover, it would be interesting to add another panel to Fig 11 that tests the anomaly of a state variable that is specifically designed to be quite poor as spiciness-as-a-state-variable — but nonetheless may appear comparably good as spiciness-as-a-property (anomaly).

In addition to the question of which geographic data should enter the construction of the $\xi_r$ function, the question of how this data is used must also be asked. Early on in the paper, the author describes this as the isopycnal mean, which presumably

implies an arithmetic mean (this should be clarified). However, Section 4 seems to make this more general, stating only that $\xi_r(\gamma)$ is a "suitably constructed function of density only". Should we use an arithmetic mean? If so, why? If we define $\xi_r$ as the best such function, in some kind of a least-squares sense, would we discover that it is an arithmetic mean? Fig 8 provides a trivial example where $\gamma$ and $\xi$ are linear functions of space. Obviously, the real ocean presents a far more nonlinear problem, for there will not be a suitable function $\xi_r(\gamma)$ that renders $\nabla\xi'$ orthogonal to $\nabla\gamma$. Unless this general issue can be addressed, Section 4.1 is not of great theoretical or practical interest.

Section 3 provides one way (among many) to nonlinearly scale the $S-\Theta$ diagram so that both axes have common units [density], such that there is a well-defined spiciness variable $\tau_{\ddagger}$ that is orthogonal to density on this diagram. However, $\tau_{\ddagger}$ is subsequently dropped from the manuscript. It is claimed (line 168) that $\tau_{\ddagger}$ is similar to $\tau_{jmd}$, but this is not proven or shown numerically. This manuscript would be considerably stronger if $\tau_{\ddagger}$ were tested in Section 4.2 *and* shown to have some advantage over other spiciness/spicity variables (including $\tau_{jmd}$, which it may well turn out to be very similar to). Otherwise, Section 3 seems to be of limited utility. The theoretical argument, opening Section 3, reaches the conclusion that the $S-\Theta$ axes should be rescaled to have density as their common units, but this is commonly known. Huang et al (2018) pursues this, for example. The author does not make it clear why this rescaling of the $S-\Theta$ diagram is superior to other rescalings, even linear ones.

In terms of structure, Section 4.2 "Illustrations" is more of a "Results" section, and does not fit well with the theoretical Section 4.1. I recommend splitting Section 4 into two sections, and expanding both, as described above.

The author claims that the anomaly $\xi'$ "is the variable optimally suited for characterising ocean water masses" (line 4-5). However, this is not proven, nor is there any discussion about how such optimality would be measured. Claims of optimality appear in several other places in the manuscript. I recommend this loose language be qualified and proved, or else changed.

Another one of the major results claimed is that this paper presents a "rigorous and first-principles theoretical justification for... a globally-defined material density variable $\gamma(S, \theta)$ maximising neutrality" (e.g. lines 10, 115-116, 251-252). However, this justification is predicated upon the desire of oceanographers to have a spiciness variable. Though such a variable may be useful to possess, it does not itself have a rigorous and first-principles theoretical foundation, and so cannot be leveraged to justify such a $\gamma$.

**Specific Comments**

29: Another citation for thermobaric instability would be apt, here, such as Ingersoll (2005; JPO).

71: Some additional conditions are necessary to make this example true. As counter-example, take $\gamma(S, \theta)$ and $\xi(S, \theta)$ as constants: both are material, but the given $d$ does not satisfy property 2, since two distinct points $(S_1, \theta_1)$ and $(S_2, \theta_2)$ would nonetheless have $d = 0$.

80: Please provide further detail on the derivation of $\gamma_S S' + \gamma_\theta \theta' \approx 0$. Is one supposed to take the gradient of $\gamma(S - S', \theta - \theta') = \gamma_0$ in the neutral tangent plane, and assume that $\gamma$ is an approximately neutral density variable? This would lead to $\gamma_S \nabla_n S' + \gamma_\theta \nabla_n \theta' \approx 0$, but this differs from the stated equation by the presence of gradients. It is not clear whether the condition $\gamma(S_r(\gamma_0), \theta_r(\gamma_0)) = \gamma_0$ is necessary "for all $\gamma_0$", or just the $\gamma_0$ under current consideration.

93-94: Fig 2 does not show, as stated, that "the ability of a variable to characterise water masses is proportional to the degree of orthogonality between $\nabla \xi$ and $\nabla \gamma$...". It simply shows that the spatial gradient of different candidate spiciness variables make different angles with $\nabla \gamma$. Fig 2 can only be interpreted as the author desires by referencing the interpretation of Fig. 1, that $S_A$ is a better spiciness variable than the other two. Even still, this is merely an interpretation or a "suggestion" at this stage.

Eq. (1): Please provide some details on the derivation of this equation. Tailleux (2016a) also lacks such details.

Eq. (4): $d_i$ needs to be defined. Also, it needs to be stated that this assumes $\gamma$ is a perfectly neutral density variable, rather than "on any given density surface..."

Eq. (6): Units error in the middle expression. $X$ and $Y$ have units of density, so cannot be added to the unitless value 1, which should be $\rho_{00}$.

Eq. (11): This isn't really the total differential of $\tau_{\ddagger}$ if it's at fixed pressure.

165: $\tau$ has not been defined. All that can be said is that $\tau_0$ is an arbitrary constant with units of density, and that $\tau_{\ddagger}(S_0, \theta_0, p) = \tau_0$.

168: $\tau_{\ddagger}$ is the exact solution to an approximate differential equation, but this does not mean $\tau_{\ddagger}$ is an approximate solution of the exact differential equation. Here, Eq. (11) is the "approximate differential equation", which approximately matches (exactly in form, approximately in coefficients) with the "exact differential equation" set out by Jackett and McDougall (1985). If this logic were true, chaos (theory) would not exist.

Eq. (14): How did $\rho_{00}$ become $\rho_0$? I assume the neutral relation $\nabla_i\theta = \alpha(S,\theta,p)\beta(S,\theta,p)^{-1}\nabla_i S$ was used, but this provides the second equality in (14) only if $\rho_0 = \rho_{00}$.

196-7: Here, the author states that Section 3 showed spiciness can be theoretically justified to be orthogonal to density in thermohaline space, but elsewhere (e.g. line 208) stated that orthogonality in thermohaline space is "fundamentally ill-defined". This is confusing, to say the least. I remain unconvinced that Section 3 delivered what has been advertised here (line 196-7). Rather, Section 3 just showed that we can define an alternative, but only approximate, equation of state under which orthogonality in thermodynamic space is well-defined. This does not answer the theoretical questions surrounding spiciness in the real ocean.

204 and Eq. (15): This is introduced a bit sloppily. No definition is given for $\hat{f}$, so the

reader is left to figure that out by understanding Eq. (15) and/or by comparison with $\hat{\rho}$ earlier. Also, $\partial \hat{f}/\partial p = \partial f/\partial p$ is used but not stated in Eq. (15), which would probably benefit by using the latter in the middle expression. Actually, since the same thing appears in Eq. (16), it may be better to simply provide an equation that does nothing more than *define* $\tilde{\nabla}$, thereby eliminating these multi-part equations (15) and (16).

206: "efficient" does not seem like the right word here. Maybe "compact"?

230: What is meant by "the values of $\sigma_1$ contours retained in the nonlinear regression"? Is only some of the data shown in Fig 9 actually used in the nonlinear regression that produces its red lines? And the data that is used has $\sigma_1$ values between the largest and smallest of the thick black contours in Fig 10? The caption of Fig 10 helps support this interpretation, but even there it is confusing: the restricted range of $\sigma_1$ used to compute the nonlinear regression should be defined by two $\sigma_1$ values (a lower and upper bound) rather than four values (the thick contours).

250: Jackett and McDougall (1985) should be cited here.

266-7: What would happen if you used a non-constant reference pressure for $\tau_{\ddagger}$, as suggested here? Actually, it's not clear what this even means: where does a reference pressure fit into $\tau_{\ddagger}$?

270: This claim, that Tailleux (2016b)'s density variable "maximizes neutrality while also being the only one that accounting for thermobaricity", is unfounded. Tailleux (2016b) only compared the neutrality of his density variable against a select few competitor density variables, namely two potential density variables, $\gamma^n$ of Jackett and McDougall (1997), and a rational approximation of $\gamma^n$ defined by McDougall and Jackett (2005; JMR). Conspicuously missing is the orthobaric density of de Szoeke et al (2000), not to mention the neutral density of Eden and Willebrand (1999). Moreover, since Tailleux (2016b)'s density variable was custom-built to mimic $\gamma^n$ of Jackett and Mc-Dougall (1997), and the latter exhibits better neutrality (Fig 6 of Tailleux (2016b)), it is unclear how the author can make this claim even if orthobaric density had been tested.

275-6: The author has not shown that $\xi'$ appears to be insensitive to the particular choice of $\xi_r(\gamma)$, since only one method for empirically constructing $\xi_r(\gamma)$ was tested, namely the (arithmetic?) mean.

278: Isn't $\xi'$ conservative by definition? Since $\xi$ and $\gamma$ are assumed to be conservative throughout this manuscript, then $\xi'$ should be too.

Fig 2: The source data should be restricted to be between, say, 500 and 1500 dbar, to remain near the reference pressure of $\sigma_1$.

Fig 2: The colors are a bit confusing. In the caption, spiciness and spicity are described as brown and orange, respectively – quite similar colors! This seems (to me) to describe more how they appear in the histogram when blended with other colors, not how they are in the legend.

Fig 9: It is nearly impossible to get much information from these panels. It is likely that most of what we see is due to outliers, and the vast majority of the data is lying on top of itself. Instead of a simple scatter plot, I suggest using a 2D histogram.

Fig 11: The colorbars all range between -2 and 2, but the units vary across panels. It would be better to let each colorbar cover the entire range of its variable, or perhaps to cover the variable's range up to two standard deviations, say.

Fig 11: Caption: Which contours of $\sigma_1$ are shown in white?

**Technical Corrections**

4-8: "The key results are:" should be "The key results are as follows." and each key result that follows should be a separate sentence. (What comes before a colon must be a complete sentence.)

9-10: Same issue as above.

19: behaves -> behave

28: sopycnal -> isopycnal

48 & 53: At this stage, it's unclear why or when "potential" should appear before "spicity" and "spiciness'.

50: remove "in general"

56, 277, 278: question mark should be a period, or rephrase so that a question is actually asked, rather than stating what the question is.

61: signal -> signals

67: The statement "checked in any good mathematics textbook" is rather cavalier, and would be better omitted. Simply naming the mathematical object $d$ as a metric is enough.

69: Using "1" and "2" to identify data leads to the unfortunate notation of $d(1,2)$. I'd suggest using $A$ and $B$ instead of numbers.

72: The definition of $f_i$ is quite confusingly written, since $(\gamma, \xi)$ is really meant to say "$\gamma$ or $\xi$".

105: This is usually called the "dianeutral vector" not the "neutral vector".

120: join -> joint

125: typo in the inline equation: the first $S$ should be $\theta$.

125: $J$ has already been defined and does not need to be stated again.

Eq (14) and line 191: $\tau$ should be $\tau_\ddagger$.

192: brackets -> braces in Eq. (14).

207: tilde is placed incorrectly, should be over $\nabla$.

259: all -> are all

260: "the one used in this study": it's unclear what "one" is referring to, since four candidates were tested, and the author's own variable $\tau_\ddagger$ was also presented.

263: "as the ... variable" -> "as ... variables"

264: mimic -> mimics

Fig 2: The x axis label is missing two gradient symbols, in front of $\sigma_1$ and $\xi$. Also, "11" -> "1". Also, "less" -> "least".

Fig 9: "Fig. 11" -> "Fig. 1". Also, shouldn't "spiciness" and "spicity" be changed to "potential spiciness" and "potential spicity" throughout this caption? Also, the subscript for $\tau_\ddagger$ is sideways on the y-axis label of panel (a).

various: showed -> shown

---

## Referee Comment (RC3) · Jan Zika (Referee) · 4 Jul 2020

Jan Zika (Referee)

j.zika@unsw.edu.au

Received and published: 4 July 2020

Tailleux presents new ideas around spiciness in the ocean. I think this is a worthwhile paper with some interesting points being made.

A number of the key conclusions don't seem well supported though. Some points are presented as self-evident, yet their justification seems far from obvious. Furthermore, some analysis lacks rigour. I feel these are largely matters of presentation and I expect I will be able to recommend publication after major revision.

Specific issues:

Orthogonality

Tailleux argues that the most appropriate spice variable should be orthogonal in geographical coordinates. I actually think this is a very important point but words like orthogonal and optimal are used frequently without their implementation actually being globally orthogonal, nor evidently 'optimal' in any way.

Firstly, the importance of orthogonality is introduced with "As is well known, the most efficient way to represent a vector is achieved by decomposing it in an orthogonal basis" This statement (and similar statements about orthogonality) should be made more precisely. For example, does the word 'efficient' have a precise meaning here?

If we are to apply rigour to the idea of developing an orthogonal basis, surely there is a fundamental issue that the gradient of any spice variable can vanish on an isopycnal (and clearly the along-section isopycnal gradient of all the spice variables shown in figure 11 vanish at various locations).

The problem Tailleux is dealing with is in three-dimensional space yet neutral density and spice offer only two basis vectors. This should be clarified with regard to the motivation to have an orthogonal basis since the basis developed is clearly incomplete.

I suggest a severe tone down of the language of 'orthogonal coordinates' unless these issues are to be discussed carefully.

Perhaps more crucially, it is unclear where and to what degree the modified spice variable eta' is actually orthogonal. How de we know if the reference profile eta\_r(\sigma\_1) is 'suitably constructed'? Fig.11 uses a polynomial fit of eta(sigma) for a specific section for eta\_r. Doesn't this imply there is no perfect orthogonality anywhere? Why not choose eta\_r to be eta at a specific latitude and longitude so at least local orthogonality is ensured? Or one could use the global isopycnal average of eta. Why not these other choices?

More generally, there is no attempt to quantify how 'optimal' different methods for making eta' orthogonal are despite it the word optimal being used frequently throughout the
**paper.**

**Fig 11.**

I think the variables shown in Fig. 11 are even closer than they appear. Both potential spiciness and spicity are in units of kg/m3 while Theta is in oC and S is in g/kg. As a consequence Theta-Theta\_r is saturated and S-S\_r is poorly resolved by the colour scale. There seems to not be a fundamental reason to care about the units of any of these coordinates since their utility is primarily in tracing water masses. So, I strongly encourage the author to rescale the colour axes (e.g. by dividing each by 1 standard deviation) so the variations in each variable are highlighted rather than their absolute values. This will likely show that all four variables look very similar in terms of their relative variations.

I am not sure if I saw it mentioned but it would be nice to see it pointed out that if the equation of state is indeed linear then all four of the diagnostics shown in Fig.11 should be proportional (at least I am sure this is the case for Theta and S).

**General references to previous work**

There are a lot of instances where what is written in previous work is generalised. These need to be either removed or replaced with concrete examples. For example on line 120 it says "So far, studies that have pursued orthogonality...have taken for granted...". Unless complete knowledge of all such studies can be claimed, it would be more appropriate to just point out that this has happened in some studies and provide references.

Other comments and suggestions:

There were a large number of typos and a few terms left un-defined.

Generally, it makes more sense to me that 'l' is used instead of 'we' since this is a sole author paper.
A lot of the mathematics was difficult to follow often because basic variables and notation were not defined.

Line 14: What is a 'binary fluid' Line 25: What is "de-compensate" Line 28: "sopycnal" Line 45: "As \*shown\* in this paper" Line 72: I think I understand that f can be either gamma or eta. But as written it looks like f maps from Theta and S into gamma and eta space (e.g. the author writes f = (gamma,eta)). This whole paragraph could be expanded for clarity as it is important. Line 80: What is gamma\_S? The partial derivative of gamma with respect to S? Line 102: "As shown by" or "As Tailleux (2016a) showed" Eq 2: Define rho\_p and rho\_eta Line 120: "in a join\*t\* system". Also – its not clear what a 'joint system of physical units' is. Eq (5): Why no brackets around what is being logged here? Line 139: Why rho\_00 and not just rho\_0? Line 177: Define 'quasi-material'

Personal note: In our recent paper, Zika, J. D., J-B. Sallée, A. C. Naveira-Garabato, A. J. Watson, A. Meijers, M-J. Messias, B. King, 2020: Tracking the spread of a passive tracer through Southern Ocean water masses. Ocean Science., 16, 323–336, 2020, we attempted to construct a coordinate which was locally orthogonal to the along isopycnal direction and also materially conserved. The coordinate was essentially S-S\_r. We chose S-S\_r because it was simpler to define than spice. Fig. 11 of this paper suggests this was a reasonable choice.

Our salinity anomaly variable was used to help understand the ispoycnal spreading of a passive tracer. There are likely other examples of work that benefited from, or would have benefitted from, such 'spicy' coordinates. I feel this paper would be better motivated if more references were made to such studies.

Sincerely Jan Zika

---

## Author Comment (AC1) · 11 Aug 2020

**Response to Referee 1**

This manuscript aims at clarifying the long-debated definition of a passive variable along neutral/isopycnal layers, commonly referred as "spiciness". The paper clarifies and demonstrates that the use of thermohaline anomalies (in particular absolute salinity) along neutral surfaces is sufficient to provide orthogonality in physical space. The long sought orthogonality in thermohaline space is showed to be flawed and not necessary to construct an inert variable along neutral surfaces. Moreover, the author discusses and resolves several issues raised by the definition of a physical variable satisfying the properties of spiciness. The existence of neutral surfaces is revealed to

be key to the construction of a spiciness-like variable. By using theoretical arguments and a quasi-linear transformation of T/S space, the author also compares published definitions based on different assumptions and unifies them under basic principles.

I found the manuscript very interesting and well written. It surely provides an important step forward to the study of water mass. I therefore only have a few minor comments and recommend this paper to be published.

**Response and suggested changes** I thank the referee for his/her supportive comments. In addition to implementing the corrections suggested, I plan on making the following changes in response.

- I will be more specific about what I mean by orthogonality in physical space in the introduction;

- I will add a quantification of the orthogonality of $\Theta$, and will also quantify the improvement in orthogonality between $\xi$ and $\xi - \xi_r(\gamma)$

- I'll try to see whether it is possible to reach definitive conclusions about whether $S'_A$ can be expected to be more dynamically inert where $\beta$ is the smallest.

**Response to specific comments**

- l60 and Fig 1 : What is the source of the data shown?
  Good point. I used the WOCE dataset, available at:
  http://icdc.cen.uni-hamburg.de/1/daten/index.php?id=woce&L=1

- Fig2 : In caption : $\sin(\nabla\sigma_1, \nabla\xi)$ ? Why is the yellow histogram closer to 0 (ie sine closer to 1, angle closer to $\pi/2$), but described as the less orthogonal? Have the

blue and yellow histograms been swapped? How does $\Theta$ variable compare to $S_A$ in terms of orthogonality?

The caption is as suggested by the referee. It is our impression that we correctly describe salinity as the variable the most orthogonal to $\sigma_1$ and can't quite reconcile what the referee says and what we say. $\Theta$ is sizeably much less orthogonal than $S_A$, I will add $\Theta$ to the histogram in the revised version.

- l149 : What would be the proportion of the world ocean covered in that range? To identify regions where a spiciness definition would be challenged could be an interesting add to the paper.

  I have not attempted to quantify the accuracy of the quasi-linear approximation of density for the ocean's water masses, as it does not really matter for the arguments developed in the paper. My main aim was to construct a variable that can be used as a proxy for the spiciness variables of Jackett and McDougall (1995) and McDougall and Krzysik (2015) extending such variables to a wider range of reference pressures, allowing among other things to use a reference pressure $p_r(S, \theta)$ or $p_r(S_A, \Theta)$.

- Along the manuscript, it is commonly referred to "orthogonality in physical space" and I think it would be nice to have a clear definition of what it means in introduction. I agree as I can see from the other comments that not doing so has created some confusion. This will be fixed in the revised version of the paper.

- I have the feeling that in regions of the ocean with temperature-driven density, salinity anomalies will have be a better choice to construct an inert variable. Am I speculating too much? Would $\Theta'$ be any better than $S'_A$ where density is salinity-driven (eg, coastal ocean, near sea-ice, Mediterranean, Red, Black Seas, . . .)? This is an interesting question, which I find difficult to answer. Indeed, as first showed by Jackett and McDougall (1995), all thermodynamic variables are approximately dynamically inert on a material approximately neutral $\gamma = constant$

density surface. Whether the approximation is better for $S'_A$ than $\Theta'$, and whether this can be proven, is an interesting suggestion that I need to think more about.

---

## Author Comment (AC2) · 11 Aug 2020

**Response to Referee 2**

**General Comments** This manuscript aims to clarify the theoretical foundation for a spiciness variable sought by many oceanographers. The main idea is that, before considering spiciness, one must first construct a good neutral density variable that is materially conserved, and then most any materially conserved function can be used to construct a spiciness variable, simply by constructing its anomaly along neutral surfaces. The author also clarifies that pursuing orthogonality of spiciness and neutral density in $S - \Theta$ space is misguided, and instead that the goal should be orthogonality of their gradients in physical space. Unfortunately, many of the advances of this paper

are overstated, either lacking justification, detail, or novelty. There are several logical errors as well. These are discussed below. I believe this manuscript has the potential to nicely tie together the theory of spiciness variables, but Major Revisions are required to get there.

**Response and proposed changes** I thank the referee for his/her careful and comprehensive review, as well as for some thought provoking comments. In addition to addressing the numerous specific suggestions as detailed below, I suggest to implement the following main changes to address his/her most important comments.

1. Rephrasing the abstract somewhat and link the present work with the previous work by Jackett and McDougall (1985). Throughout the manuscript, articulate better the similarities and differences with JM85 and include citation and discussion of McDougall and Giles (1987).

2. Improve the discussion of how to construct $\xi_r(\gamma)$, including a discussion of global versus local considerations

3. Improve the discussion of what is exactly meant by orthogonality in physical space and improve its theoretical justification.

4. Rephrase the various parts that appear to be causes of confusion.

5. Clarify what is meant by 'optimality'

6. Clarify what is meant by first principles justification of neutrality

In the following, I provide some more detailed responses. I did not respond to the more technical comments, which will be done when invited to revise my paper.

- One of the major points of the paper, that what matters for spiciness is actually the neutral density variable, was made by Jacket and McDougall (1985). The author has acknowledged this in some places, but a reader could easily get the impression that this idea owes to this manuscript. A stand-out example is in the abstract (line 5): stating "contrary to what is usually assumed" is unfair. Anyone who has read Jackett and McDougall (1985) would not assume this. This phrase should be removed, and a citation to Jackett and McDougall (1985) given in the abstract.

Since the referee emphasises rigour and accuracy throughout his/her review, it seems important to point out that my point is actually that what determines the dynamical inertness of spiciness is the degree of neutrality of $\gamma$, which is quite different from what the referee says. Moreover, last time I checked, the term 'neutral density' is nowhere mentioned in Jackett and McDougall (1985) (JM85). Rather, JM85 use the term 'isopycnal surfaces', stating at the beginning of their paper: "In this paper we approximate the isopycnal surfaces by surfaces of constant potential density which we call $\rho$." Moreover, JM85 also state: "the variations of any variable, when measured along isopycnal surfaces, are dynamically passive [...]". Since such a statement is technically true only on surfaces of constant in-situ density, there is no logical reason why readers would conclude that by 'isopycnal surfaces' JM85 actually mean 'neutral surfaces', especially as neither neutral surfaces nor patched potential density surfaces where yet in widespread use at the time. Even when such surfaces started to become popular, I am not aware that they have never associated with the dynamical inertness of spiciness before my paper.

I don't understand how my "contrary to what is usually assumed" is unfair to JM85, since what such a statement criticises is past studies contending that orthogonality and dynamical inertness are somehow connected, e.g., Veronis (1972). This is obviously not the case of JM85, which my paper praises for being the first to recognise that orthogonality is not connected with dynamical inertness. This being said, I agree that JM85 deserve a place in the abstract, not for promoting the use of neutral density, but for being the first to recognise the lack of connection between orthogonality and dynamical inertness, as well as for being the first to propose describing spiciness in terms of the isopycnal anomaly of some thermodynamic variable. As regards to the latter point, I don't think I give enough credit to JM85, which I plan on correcting in revising my paper.

As a final remark, I'd like to stress that I am not claiming credit for the idea that spiciness should be used in conjunction with neutral density, because there is no ambiguity that this is implied explicitly or implicitly in most if not all spicity/spiciness papers written post JM85, as is evident in Huang (2011) or McDougall and Krzysik (2015) for instance. I only claim credit for spelling a logical and rigorous argument establishing the link between dynamical inertness and neutrality, which had never been explicitly stated before, as far as I know.

- I find Fig 11 the most interesting aspect of this work. It is essentially a global test of the Jackett and McDougall (1985) idea, repeated here, that it is the anomaly $\xi'$ is dynamically inert. The author's anomaly is defined as relative to a global isopycnal average. The results are evidently meaningful, but it is not entirely clear that more refined results could be obtained by refining the averaging procedure. McDougall and Giles (1987) argued in favor of studying property (salinity) anomalies relative to a local isopycnal average. To study a particular water mass intrusion, the state of the ocean far away should be irrelevant. It would therefore be prudent of the author to discuss the utility of using global isopycnal averages, and to locate the present work relative to the earlier work of McDougall and Giles (1987).

I fully agree with the referee and thank him/her for pointing out the highly relevant study by McDougall and Giles (1987), which I was not aware of. I also agree that local versus global definitions of of $\xi_r(\gamma)$ need to be discussed in the revised version of my paper.

- Moreover, it would be interesting to add another panel to Fig 11 that tests the

anomaly of a state variable that is specifically designed to be quite poor as spiciness-as-a-state variable — but nonetheless may appear comparably good as spiciness-as-a-property (anomaly).

I am not entirely sure how to construct such a variable or how useful that would be, since it seems to me that $\Theta$ is naturally a poor choice of spiciness-as-a-state variable, which performs much better as an isopycnal anomaly. I therefore don't plan on pursuing the referee's suggestion.

- In addition to the question of which geographic data should enter the construction of the $\xi_r(\gamma)$ function, the question of how this data is used must also be asked. Early on in the paper, the author describes this as the isopycnal mean, which presumably implies an arithmetic mean (this should be clarified). However, Section 4 seems to make this more general, stating only that $\xi_r(\gamma)$ is a "suitably constructed function of density only". Should we use an arithmetic mean? If so, why? If we define $\xi_r$ as the best such function, in some kind of a least-squares sense, would we discover that it is an arithmetic mean? Fig 8 provides a trivial example where $\gamma$ and $\xi$ are linear functions of space. Obviously, the real ocean presents a far more nonlinear problem, for there will not be a suitable function $\xi_r(\gamma)$ that renders $\nabla\xi'$ orthogonal to $\nabla\gamma$. Unless this general issue can be addressed, Section 4.1 is not of great theoretical or practical interest.

Until recently, I must confess that I had no very definite ideas about the best way to construct $\xi_r(\gamma)$ or whether such a best way existed, which is what I had left the question as an open question in my discussion/conclusions. The term 'suitably constructed' was meant to leave it up to the reader to choose what they think best for their particular application, my hope being that referees more expert than I be would pick up on it and perhaps give me some pointers. However, upon giving the matter further thought, it is becoming clearer to me that in order for my framework to be self-consistent, there is only one logical way to construct $\xi_r(\gamma)$, namely as the function that maximises the orthogonality of $\nabla\gamma$ and $\nabla(\xi - \xi_r(\gamma))$.

Although there is no unique way to construct a cost function, one that is natural and easy to minimise analytically is:

$$E = \int_V [\nabla\gamma \cdot \nabla(\xi - \xi_r(\gamma))]^2 \, \mathrm{d}V = \int_V [\nabla\gamma \cdot \nabla\xi - \xi_r'(\gamma)|\nabla\gamma|^2]^2 \, \mathrm{d}V \qquad (1)$$

Minimising such a cost function with respect to $\xi_r'(\gamma)$ leads one to define the latter as an exact solution of the following problem:

$$\int_V |\nabla\gamma|^2 \left( \nabla\gamma \cdot \nabla\xi - \xi_r'(\gamma)|\nabla\gamma|^2 \right) \, \mathrm{d}V = 0. \qquad (2)$$

In the case where exact orthogonality can be enforced, as in the idealised case depicted in Fig. 8, $\xi_r'(\gamma)$ is defined by

$$\xi_r'(\gamma) = \frac{\nabla\gamma \cdot \nabla\xi}{|\nabla\gamma|^2} \qquad (3)$$

The above problem defines $\xi_r(\gamma)$ up to an integration constant, which one may fix by imposing that the resulting $\xi_r(\gamma)$ minimises the integral $\int_V |\xi - \xi_r(\gamma)| \, \mathrm{d}V$. As a result, I don't think that the current way I have constructed $\xi_r(\gamma)$ in my paper, namely in terms of a smoothing interpolating polynomial, is well justified. I am currently exploring the possibility of constructing $\xi_r(\gamma)$ as per the method outlined above, and propose to revise my paper accordingly.

- Section 3 provides one way (among many) to nonlinearly scale the $S-\Theta$ diagram so that both axes have common units [density], such that there is a well-defined spiciness variable $\tau_\ddagger$ that is orthogonal to density on this diagram. However, $\tau_\ddagger$ is subsequently dropped from the manuscript. It is claimed (line 168) that $\tau_\ddagger$ is similar to $\tau_{jmd}$, but this is not proven or shown numerically. This manuscript would be considerably stronger if $\tau_\ddagger$ were tested in Section 4.2 and shown to have some advantage over other spiciness/ spicity variables (including $\tau_{jmd}$, which it

may well turn out to be very similar to). Otherwise, Section 3 seems to be of limited utility.

*I am puzzled by this comment, because the similarity of $\tau_\ddagger$ to $\tau_{jmd}$ is established in Figs. 5 and 6, as well as indirectly in Fig. 7. I did not use $\tau_\ddagger$ further in my paper because oceanographic sections obtained with it were indistinguishable from those obtained with $\tau_{jmd}$. The primary advantages of $\tau_\ddagger$ over $\tau_{jmd}$, apart from its associated Jacobian being non-zero over all of $(S, \theta)$ space unlike that for $\tau_{jmd}$, are practical and due to its continuous dependence on pressure and having an exact mathematical form in terms of in-situ density that clarifies its dependence on its arbitrary tunable parameters. Since availabele software for $\tau_{jmd}$ is only limited to a couple of reference pressures, this makes $\tau_\ddagger$ considerably more flexible to use in practice.*

- The theoretical argument, opening Section 3, reaches the conclusion that the $S - \Theta$ axes should be rescaled to have density as their common units, but this is commonly known. Huang et al (2018) pursues this, for example. The author does not make it clear why this rescaling of the $S - \Theta$ diagram is superior to other rescalings, even linear ones.

  *I am puzzled by this comment because my argument is actually the other way around. Indeed, the starting point of my argument explicitly recognises that many investigators have sought to re-scale $S - \Theta$ in density coordinates, however without really justifying it other than by invoking naturalness or convenience. This is expressed Line 120-121 by "So far, studies that have pursued orthogonality in one form or the other have taken it for granted that such a joint system of units should be based on density unit, but without really proving it." What Section 3 explores is whether it is possible to make a stronger and more rigorous case for a re-scaling in density units. I argue that this is possible, the key being to remark that (Lines 122-123 and Equation 4) that "the isopycnal variations of any material*

*function $\xi(S, \theta)$ on any given density surface $\gamma(S, \theta) = $ constant satisfy"*

$$\mathrm{d}_i\xi = \frac{J}{\gamma_S\gamma_\theta}\gamma_S\mathrm{d}_iS = -\frac{J}{\gamma_S\gamma_\theta}\gamma_\theta\mathrm{d}_i\theta$$

*(my equation 4). This result is important because it states that the isopycnal variations of $\xi$ are all proportional to the quantity $\gamma_S\mathrm{d}_iS = -\gamma_\theta\mathrm{d}_i\theta$, which has density units, regardless of $\xi$.*

*The only point I am trying to make in this section that it is possible to regard the construction of 'spicity'-like and spiceness-like variables as fundamentally relying on similar theoretical foundations, since the two can be interpreted as being orthogonal to density in re-scaled $(X(S), Y(\theta))$ coordinates, the former for a linear rescaling, the latter for a nonlinear one. The referee may be right to think that there is therefore no particular advantage of the nonlinear re-scaling over the linear re-scaling, since $X(S)$ and $Y(\theta)$ rely on the specification of arbitrary constants regardless of how they are constructed, suggesting that spiceness is as arbitrary as spicity in some sense. So far, however, Jackett and McDougall (1985), Flament (2002) or McDougall and Krzysik (2015) have argued that spiceness-like variables are superior to spicity-like variables, even if the physical basis for their arguments is not crystal clear, to say the least.*

- In terms of structure, Section 4.2 "Illustrations" is more of a "Results" section, and does not fit well with the theoretical Section 4.1. I recommend splitting Section 4 into two sections, and expanding both, as described above.

  *I need more time to ponder about to restructure my paper, as I also need to account for the other referees' remarks.*

- The author claims that the anomaly $\xi'$ "is the variable optimally suited for characterising ocean water masses" (line 4-5). However, this is not proven, nor is there any discussion about how such optimality would be measured. Claims of optimality appear in several other places in the manuscript. I recommend this loose

language be qualified and proved, or else changed.
I agree with this comment. I need to ponder how to best address it in my revision.

- Another one of the major results claimed is that this paper presents a "rigorous and first-principles theoretical justification for... a globally-defined material density variable $\gamma(S,\theta)$ maximising neutrality" (e.g. lines 10, 115-116, 251-252). However, this justification is predicated upon the desire of oceanographers to have a spiciness variable. Though such a variable may be useful to possess, it does not itself have a rigorous and first-principles theoretical foundation, and so cannot be leveraged to justify such a $\gamma$.
I don't think it is true that my theoretical justification for $\gamma$ is "predicated upon the desire of oceanographers to have a spiciness variable"; nevertheless, I need to ponder somewhat more about the referee's view to establish whether I agree with it or not. I'll try to address the issue explicitly in my revision.

**Specific Comments**

- 29: Another citation for thermobaric instability would be apt, here, such as Ingersoll (2005; JPO).
I agree that this is a good reference to cite

- 71: Some additional conditions are necessary to make this example true. As counterexample, take $\gamma(S,\theta)$ and $\xi(S,\theta)$ as constants: both are material, but the given $d$ does not satisfy property 2, since two distinct points $(S_1, \theta_1)$ and $(S_2, \theta_2)$ would nonetheless have $d = 0$.
I think the counterexample is not in the spirit of the paper

- 80: Please provide further detail on the derivation of $\gamma_S S' + \gamma_\theta \theta' \approx 0$. Is one supposed to take the gradient of $\gamma(S - S', \theta - \theta') = \gamma_0$ in the neutral tangent plane, and assume that $\gamma$ is an approximately neutral density variable? This

would lead to $\gamma_S \nabla_n S' + \gamma_\theta \nabla_n \theta' \approx 0$, but this differs from the stated equation by the presence of gradients. It is not clear whether the condiiton $\gamma(S_r(\gamma_0), \theta_r(\gamma_0)) = \gamma_0$ is necessary "for all $\gamma_0$", or just the $\gamma_0$ under consideration.
I will do so in the revision

- 93-94: Fig 2 does not show, as stated, that "the ability of a variable to characterise water masses is proportional to the degree of orthogonality between $\nabla \xi$ and $\nabla \gamma$ ..." It simply shows that the spatial gradient of different candidate spiciness variables make different angles with $\nabla \gamma$. . Fig 2 can only be interpreted as the author desires by referencing the interpretation of Fig. 1, that SA is a better spiciness variable than the other two. Even still, this is merely an interpretation or a "suggestion" at this stage.
Fair enough

- Eq. (1): Please provide some details on the derivation of this equation. Tailleux (2016a) also lacks such details.
This equation is obtained by the Jacobi method. It is explained in much details in Appendix A of Feistel (2018): Thermodynamic properties of seawater, ice and humid air: TEOS-10, before and beyond. Ocean Sciences, 14, 471–502.
`https://os.copernicus.org/articles/14/471/2018/os-14-471-2018.pc`

- Eq. (4): $d_i$ needs to be defined. Also, it needs to be stated that this assumes is a perfectly neutral density variable, rather than "on any given density surface..."
The expression is correct as stated. $d_i$ is defined by introducin the isopycna/diapycnal decomposition $\nabla F = \nabla_i F + \nabla_d F$ for any scalar function $F$, in which case we have $dF = \nabla F \cdot dx$ and $d_i F = \nabla_i F \cdot dx$. I am a little bit surprised that this is needed.

- Eq. (6): Units error in the middle expression. X and Y have units of density, so cannot be added to the unitless value 1, which should be $\rho_{00}$.

- Eq. (11): This isn't really the total differential of $\tau_{\ddagger}$ if it's at fixed pressure.

- 165: $\tau$ has not been defined. All that can be said is that $\tau_0$ is an arbitrary constant with units of density, and that $\tau_{\ddagger}(S_0, \theta_0, p) = \tau_0$).

- 168: $\tau_{\ddagger}$ is the exact solution to an approximate differential equation, but this does not mean $\tau_{\ddagger}$ is an approximate solution of the exact differential equation. Here, Eq. (11) is the "approximate differential equation", which approximately matches (exactly in form, approximately in coefficients) with the "exact differential equation" set out by Jackett and McDougall (1985). If this logic were true, chaos (theory) would not exist.
  The referee has lost me

- Eq. (14): How did $\rho_{00}$ become $\rho_0$? I assume the neutral relation $\nabla_i \theta = \alpha(S, \theta, p)\beta(S, \theta, p)^{-1}\nabla_i S$ was used, but this provides the second equality in (14) only if $\rho_0 = \rho_{00}$.

- 196-7: Here, the author states that Section 3 showed spiciness can be theoretically justified to be orthogonal to density in thermohaline space, but elsewhere (e.g. line 208) stated that orthogonality in thermohaline space is "fundamentally ill-defined". This is confusing, to say the least. I remain unconvinced that Section 3 delivered what has been advertised here (line 196-7). Rather, Section 3 just showed that we can define an alternative, but only approximate, equation of state under which orthogonality in thermodynamic space is well-defined. This does not answer the theoretical questions surrounding spiciness in the real ocean.
  The referee misunderstands what Section 3 is about. As we all agree, spiciness is a property that is best understood as an anomaly $\xi' = \xi - \xi_r(\gamma)$ for some thermodynamic property and suitably constructed $\xi_r(\gamma)$ to be defined. The theoretical questions that surrounds spiciness in the ocean are therefore:

    1. Is there theoretical advantage or justification in using any particular form of

    $\xi$ over another one, or is one free to choose whatever $\xi$ we like? Assuming that there exists some better choice of $\xi$, should it be defined along the lines suggested by Jackett and McDougall (1985), Flament (2002), McDougall and Krzysik (2015), along the lines suggested by Veronis (1972), Huang (2011), Huang et al (2018), or along some other lines still to be discovered?

    2. Once one has settled on a particular choice of $\xi$, what is the best way to construct $\xi_r(\gamma)$ and if so, what is its theoretical justification?

  As regards to point (1), it is clear that Jackett and McDougall (1985), Flament (2002) and McDougall and Krzysik (2015) believe that there are advantages in defining $\xi'$ based on a dedicated spiciness-as-a-state function constructed along the lines that they propose. This is especially evident in the last part of Jackett and McDougall (1985), who argued that their $\tau'_{jmd}$ is better than $S'$ or $\tau'_\nu$, the latter being based on Veronis (1972) variable. By 'fundamentally ill defined' I meant that such a construction has no intrinsic physical meaning since the construction of $X(S)$ and $Y(\theta)$ involve the specification of arbitrarily defined constants regardless of how these are defined, even when these are chosen to be nonlinear.

- 204 and Eq. (15): This is introduced a bit sloppily. No definition is given for $\hat{f}$ so the reader is left to figure that out by understanding Eq. (15) and/or by comparison with $\hat{\rho}$ earlier. Also, $\partial \hat{f}/\partial p = \partial f/\partial p$ is used but not stated in Eq. (15), which would probably benefit by using the latter in the middle expression. Actually, since the same thing appears in Eq. (16), it may be better to simply provide an equation that does nothing more than define $\hat{\nabla}$, thereby eliminating these multi-part equations (15) and (16).
  I'll take the referee's comment into account when revising the paper.

- 206: "efficient" does not seem like the right word here. Maybe "compact"?
  May be

- 230: What is meant by "the values of $\sigma_1$ contours retained in the nonlinear regression"? Is only some of the data shown in Fig 9 actually used in the nonlinear regression that produces its red lines? And the data that is used has $\sigma_1$ values between the largest and smallest of the thick black contours in Fig 10? The caption of Fig 10 helps support this interpretation, but even there it is confusing: the restricted range of $\sigma_1$ used to compute the nonlinear regression should be defined by two $\sigma_1$ values (a lower and upper bound) rather than four values (the thick contours).
  I believe most readers will understand that the lower and upper bounds are given by the leftmost and rightmost contours.

- 250: Jackett and McDougall (1985) should be cited here.
  Agreed

- 266-7: What would happen if you used a non-constant reference pressure for $\tau_{\ddagger}$, as suggested here? Actually, it's not clear what this even means: where does a reference pressure fit into $\tau_{\ddagger}$?
  The idea here is to use $\tau_{\ddagger}(S, \theta, p_r(S, \theta))$, which would be the natural approach for use with $\gamma^T = \rho(S, \theta, p_r) - f(p_r)$.

- 270: This claim, that Tailleux (2016b)'s density variable "maximizes neutrality while also being the only one that accounting for thermobaricity", is unfounded. Tailleux (2016b) only compared the neutrality of his density variable against a select few competitor density variables, namely two potential density variables, $\gamma^n$ of Jackett and McDougall (1997), and a rational approximation of $\gamma^n$ defined by McDougall and Jackett (2005; JMR). Conspicuously missing is the orthobaric density of de Szoeke et al (2000), not to mention the neutral density of Eden and Willebrand (1999). Moreover, since Tailleux (2016b)'s density variable was custom-built to mimic n of Jackett and Mc- Dougall (1997), and the latter exhibits better neutrality (Fig 6 of Tailleux (2016b)), it is unclear how the author can make

this claim even if orthobaric density had been tested
Orthobaric density is not a purely material variable so does not fit in the present framework. However, McDougall and Jackett (2005) $\gamma_a$ and Eden and Willebrand (1999)'s variable are presumably affected by thermobaricity on account to the way that they are constructed, so I agree with the referee that I was imprecise.

- 275-6: The author has not shown that $\xi'$ appears to be insensitive to the particular choice of $\xi_r(\gamma)$, since only one method for empirically constructing $\xi_r(\gamma)$ was tested, namely the (arithmetic?) mean.
  The use of the verb 'appears' makes my statement a subjective one. Nevertheless, I can try to be somewhat more precise in the revision.

- 278: Isn't $\xi'$ conservative by definition? Since $\xi$ and $\gamma$ are assumed to be conservative throughout this manuscript, then $\xi'$ should be too.
  The term conservative is used as in McDougall (2003) paper on Conservative Temperature. A conservative variable $H$ is one that satisfies an equation of the form
$$\frac{DH}{Dt} + \nabla \cdot F_h = 0, \tag{4}$$
  $\gamma$ and $\xi$ are assumed to be material, not necessarily conservative. $\gamma$ certainly is not since it is affected by cabbelling and thermobaricity potentially.

- Fig 2: The source data should be restricted to be between, say, 500 and 1500 dbar, to remain near the reference pressure of $\sigma_1$.
  I disagree as the point of the figure is to show the distribution of water masses as represented by different spiciness variables.

- Fig 2: The colors are a bit confusing. In the caption, spiciness and spicity are described as brown and orange, respectively – quite similar colors! This seems (to me) to describe more how they appear in the histogram when blended with

other colors, not how they are in the legend.
I'll correct this in the revised version of the paper.

- Fig 9: It is nearly impossible to get much information from these panels. It is likely that most of what we see is due to outliers, and the vast majority of the data is lying on top of itself. Instead of a simple scatter plot, I suggest using a 2D histogram.
I don't understand what would be the purpose of what the referee suggests.

- Fig 11: The colorbars all range between -2 and 2, but the units vary across panels. It would be better to let each colorbar cover the entire range of its variable, or perhaps to cover the variable's range up to two standard deviations, say.
Dr. Zika suggested to normalise each variable by its standard deviation, which I think is a good idea to address the problem.

- Fig 11: Caption: Which contours of $\sigma_1$ are shown in white?
This will be clarified in the revised version of the paper

Technical Corrections Thank you for these. These will be taken into account while revising the paper.

- 4-8: "The key results are:" should be "The key results are as follows." and each key result that follows should be a separate sentence. (What comes before a colon must be a complete sentence.)

- 9-10: Same issue as above.

- 19: behaves -> behave

- 28: sopycnal -> isopycnal

- 48 and 53: At this stage, it's unclear why or when "potential" should appear before "spicity" and "spiciness'.

- 50: remove "in general"

- 56, 277, 278: question mark should be a period, or rephrase so that a question is actually asked, rather than stating what the question is.

- 61: signal -> signals

- 67: The statement "checked in any good mathematics textbook" is rather cavalier, and would be better omitted. Simply naming the mathematical object d as a metric is enough.

- 69: Using "1" and "2" to identify data leads to the unfortunate notation of d(1; 2). I'd suggest using A and B instead of numbers.

- 72: The definition of $f_i$ is quite confusingly written, since $(\gamma, \xi)$ is really meant to say $\gamma$ or $\xi$"

- 105: This is usually called the "dianeutral vector" not the "neutral vector".

- 120: join $\rightarrow$ joint

- 125: typo in the inline equation: the first $S$ should be $\theta$

- 125: J has already been defined and does not need to be stated again.

- Eq (14) and line 191: $\tau$ should be $\tau_{\ddagger}$ .

- 192: brackets -> braces in Eq. (14).

- 207: tilde is placed incorrectly, should be over $\nabla$

- 259: all -> are all

- 260: "the one used in this study": it's unclear what "one" is referring to, since four candidates were tested, and the author's own variable was also presented.

- 263: "as the ... variable" → "as ... variables"

- 264: mimic → mimics

- Fig 2: The x axis label is missing two gradient symbols, in front of $\sigma_1$ and $\xi$. Also, "11" -> "1". Also, "less" → "least".

- Fig 9: "Fig. 11" → "Fig. 1". Also, shouldn't "spiciness" and "spicity" be changed to "potential spiciness" and "potential spicity" throughout this caption? Also, the subscript for $\tau_\ddagger$ is sideways on the y-axis label of panel (a).

- various: showed → shown

---

## Author Comment (AC3) · 11 Aug 2020

Response to Referee 3

Tailleux presents new ideas around spiciness in the ocean. I think this is a worthwhile paper with some interesting points being made. A number of the key conclusions don't seem well supported though. Some points are presented as self-evident, yet their justification seems far from obvious. Furthermore, some analysis lacks rigour. I feel these are largely matters of presentation and I expect I will be able to recommend publication after major revision.

**Response and proposed changes** I thank Dr. Zika for his careful review and useful suggestions. I think that my analysis is rigorous enough but I agree in the light of Dr

[Figure]

Zika's comments that some of my arguments are not precise or tight enough. The main changes that I propose to implement following his suggestions, in addition to accounting for more specific suggestions, are:

1. Improve the discussion of orthogonality

2. Improve Fig. 11 by rescaling the variables by their standard deviation

3. Improve what I mean by 'optimality'

**Specific issues:**

1. **Orthogonality** Tailleux argues that the most appropriate spice variable should be orthogonal in geographical coordinates. I actually think this is a very important point but words like orthogonal and optimal are used frequently without their implementation actually being globally orthogonal, nor evidently 'optimal' in any way. Firstly, the importance of orthogonality is introduced with "As is well known, the most efficient way to represent a vector is achieved by decomposing it in an orthogonal basis" This statement (and similar statements about orthogonality) should be made more precisely. For example, does the word 'efficient' have a precise meaning here? If we are to apply rigour to the idea of developing an orthogonal basis, surely there is a fundamental issue that the gradient of any spice variable can vanish on an isopycnal (and clearly the along-section isopycnal gradient of all the spice variables shown in figure 11 vanish at various locations). The problem Tailleux is dealing with is in three-dimensional space yet neutral density and spice offer only two basis vectors. This should be clarified with regard to the motivation to have an orthogonal basis since the basis developed is clearly incomplete. I suggest a severe tone down of the language of 'orthogonal coordinates' unless these issues are to be discussed carefully. Perhaps more

crucially, it is unclear where and to what degree the modified spice variable eta'
is actually orthogonal. How de we know if the reference profile $\xi_r(\sigma_1)$ is 'suitably
constructed'? Fig.11 uses a polynomial fit of $\xi_r(\sigma_1)$ for a specific section for $\xi_r$.
Doesn't this imply there is no perfect orthogonality anywhere? Why not choose
$\eta_r$ to be $\eta$ at a specific latitude and longitude so at least local orthogonality is en-
sured? Or one could use the global isopycnal average of $\eta$ Why not these other
choices? More generally, there is no attempt to quantify how 'optimal' different
methods for making eta' orthogonal are despite it the word optimal being used
frequently throughout the paper.

Dr. Zika makes a number of legitimate points that I agree will need to be clarified
in the revised version of the paper.

- The underlying physical problem that one tries to address here is how best
  to construct a new set of coordinates $(\gamma, \xi)$ to isolate the active from the
  passive parts of $(S, \theta)$. Upon such a change of coordinates, functions of
  $f(S, \theta)$ are transformed into functions $\tilde{f}(\gamma, \xi)$. The gradient of such functions
  can be written in the following equivalent forms:

$$\nabla f = f_S \nabla S + f_\theta \nabla \theta = \tilde{f}_\gamma \nabla \gamma + \tilde{f}_\xi \nabla \xi = (\tilde{f}_\gamma + \tilde{f}_\xi \xi_r'(\gamma)) \nabla \gamma + \tilde{f}_\xi \nabla (\xi - \xi_r(\gamma)) \quad (1)$$

Physically, when $f$ is taken to be in-situ density $\rho = \tilde{\rho}(\gamma, \xi, p)$, one wants to:

  - Minimise the dependence of $\tilde{\rho}$ on $\xi$, that is make the partial derivative
    $\partial \tilde{\rho}/\partial \xi$ as small as possible. This is equivalent to make $\gamma$ as neutral as
    feasible.
  - One also would like to minimise the contribution $\tilde{\rho}_\xi \nabla(\xi - \xi_r(\gamma))$ so that
    the term proportional to $\gamma$ maximally projects on the neutral vector. To
    that end, it is easy to establish that one needs to maximise the orthog-
    onality of $\nabla \xi'$ and $\nabla \gamma$. Locally, this is possible if one define $\xi_r(\gamma)$ so
    that

$$\xi_r'(\gamma) = \frac{\nabla \xi \cdot \nabla \gamma}{|\nabla \gamma|^2} \quad (2)$$

- The above equation provides a new way of computing $\xi_r(\gamma)$ that I did not fully realise when I originally wrote the paper, but which is different from specifying it in terms of polynomial interpolation or from some other form of isopycnal mean. I propose to revise the paper to be based on the above construction, which is more logical.

- Other than that, Dr. Zika is right that exact orthogonality cannot be imposed in the most general case. In that case, all what one can do is to maximise orthogonality, rather than strictly enforce it.

- Dr Zika is also right that one may want to use a more regional construction of $\xi_r(\gamma)$. However, if one wants to be able to compare the spiciness of various parts of the global ocean, $\xi_r(\gamma)$ has to be constructed globally.

- Note that for functions $f(S,\theta)$, $\nabla f$ is generated by only two vectors $\nabla S$ and $\nabla\theta$, so that its dimensionality is two rather than 3. The fact that $\nabla f$ lives in 3D space is irrelevant. With $\nabla\gamma$ and $\nabla\xi'$, one wants to create a basis to represent $\nabla f$, not a basis for all possible three-dimensional vectors.

2. **Fig 11.** I think the variables shown in Fig. 11 are even closer than they appear. Both potential spiciness and spicity are in units of kg/mȨ̈3 while $\Theta$ is in $°C$ and S is in g/kg. As a consequence $\Theta - \Theta_r$ is saturated and $S - S_r$ is poorly resolved by the colour scale. There seems to not be a fundamental reason to care about the units of any of these coordinates since their utility is primarily in tracing water masses. So, I strongly encourage the author to rescale the colour axes (e.g. by dividing each by 1 standard deviation) so the variations in each variable are highlighted rather than their absolute values. This will likely show that all four variables look very similar in terms of their relative variations.
This is a very good idea that I will implement in the revision.

3. I am not sure if I saw it mentioned but it would be nice to see it pointed out that if the equation of state is indeed linear then all four of the diagnostics shown in

Fig.11 should be proportional (at least I am sure this is the case for Θ and S).

4. General references to previous work There are a lot of instances where what is written in previous work is generalised. These need to be either removed or replaced with concrete examples. For example on line 120 it says "So far, studies that have pursued orthogonality: : :have taken for granted: : :". Unless complete knowledge of all such studies can be claimed, it would be more appropriate to just point out that this has happened in some studies and provide references.
Thank for pointing this out. I'll endeavour to be more factual and specific in the revision.

Other comments and suggestions:
Thank for these. These will be accounted for when revising the paper

1. There were a large number of typos and a few terms left un-defined.

2. Generally, it makes more sense to me that 'I' is used instead of 'we' since this is a sole author paper.

3. A lot of the mathematics was difficult to follow often because basic variables and notation were not defined.

4. Line 14: What is a 'binary fluid'

5. Line 25: What is "de-compensate"
At the surface, a density-compensated temperature anomaly will be modified by air-sea interactions, without necessarily modifying the associated density-compensated salinity anomaly. As a result, the modified temperature anomaly can no longer be density compensated, hence the term 'de-compensate'.

6. Line 28: "isopycnal"

7. Line 45: "As *shown* in this paper" Line 72: I think I understand that f can be either gamma or eta. But as written it looks like f maps from Theta and S into gamma and eta space (e.g. the author writes $f = (\gamma, \eta)$). This whole paragraph could be expanded for clarity as it is important.

8. Line 80: What is $\gamma_S$? The partial derivative of gamma with respect to S?

9. Line 102: "As shown by" or "As Tailleux (2016a) showed" Eq 2: Define $\rho_p$ and $\rho_\eta$

10. Line 120: "in a join*t* system". Also – its not clear what a 'joint system of physical units' is. Eq (5): Why no brackets around what is being logged here?

11. Line 139: Why $\rho_{00}$ and not just $\rho_0$?

12. Line 177: Define 'quasi-material' Personal note: In our recent paper, Zika, J. D., J-B. Sallée, A. C. Naveira-Garabato, A. J. Watson, A. Meijers, M-J. Messias, B. King, 2020: Tracking the spread of a passive tracer through Southern Ocean water masses. Ocean Science.,16, 323–336, 2020, we attempted to construct a coordinate which was locally orthogonal to the along isopycnal direction and also materially conserved. The coordinate was essentially $S - S_r$. We chose $S - S_r$ because it was simpler to define than spice. Fig. 11 of this paper suggests this was a reasonable choice. Our salinity anomaly variable was used to help understand the ispoycnal spreading of a passive tracer. There are likely other examples of work that benefited from, or would have benefitted from, such 'spicy' coordinates. I feel this paper would be better motivated if more references were made to such studies.
I now remember that Dr. Zika mentioned this study to me at Ocean Sciences, and I am sorry that I forgot to cite it. This will be corrected in the revision.

---

## Author Response (AR1)

**Response to Referee 1**

This manuscript aims at clarifying the long-debated definition of a passive variable along neutral/isopycnal layers, commonly referred as "spiciness". The paper clarifies and demonstrates that the use of thermohaline anomalies (in particular absolute salinity) along neutral surfaces is sufficient to provide orthogonality in physical space. The long sought orthogonality in thermohaline space is showed to be flawed and not necessary to construct an inert variable along neutral surfaces. Moreover, the author discusses and resolves several issues raised by the definition of a physical variable satisfying the properties of spiciness. The existence of neutral surfaces is revealed to be key to the construction of a spiciness-like variable. By using theoretical arguments and a quasi-linear transformation of T/S space, the author also compares published definitions based on different assumptions and unifies them under basic principles.

I found the manuscript very interesting and well written. It surely provides an important step forward to the study of water mass. I therefore only have a few minor comments and recommend this paper to be published.

**Response and suggested changes** I thank the referee for his/her supportive comments. I response to his/her specific comments, I have made the following main changes:

- Orthogonality in physical space has now been more clearly defined. In the revised version of the paper, it is now defined as the median of all angles between  $\nabla \gamma$  and  $\nabla \xi$  estimated for all available data points.
- The orthogonality between  $\nabla \xi$  and  $\nabla \gamma$  has now been quantified for all spiciness variables  $\xi$ , as well as for their anomaly  $\xi = \xi \xi_r(\gamma)$ .

I have also made many additional changes in response to the other comments, the main ones being:

- I have replaced  $\sigma_1$  by the more neutral thermodynamic neutral density  $\gamma^T$ , for which I have proposed a new and computationally simpler implementation that should facilitate the reproducibility of my results. Section 2 has been modified to include a description of the new variable.
- To work with  $\gamma^T$ , the potential spiciness/spicity variables need to be referenced to a variable reference pressure  $p_r(S, \theta)$ . How to do that is now the topic of Section 3.
- Section 4 provides an illustration of the results as before but has been completely rewritten to account for the remarks of the referees as well as to account for the use of  $\gamma^T$  instead of  $\sigma_1$ .
- Many sections have been rewritten/rephrased in the light of the new insights achieved since the original submission.

**Response to specific comments**

- l60 and Fig 1 : What is the source of the data shown? Good point. I used the WOCE dataset, available at: http://icdc.cen.uni-hamburg.de/1/daten/index.php?id=woce&L=1 Reference has now been added to the text.
- Fig2 : In caption :  $\sin(\nabla \sigma_1, \nabla \xi)$  ? Why is the yellow histogram closer to 0 (ie sine closer to 1, angle closer to  $\pi/2$ ), but described as the less orthogonal? Have the blue and yellow histograms been swapped? How does  $\Theta$  variable compare to  $S_A$  in terms of orthogonality?

The caption is as suggested by the referee. I think salinity is correctly described as the variable the most orthogonal to  $\sigma_1$  and can't quite reconcile what the referee says and what I say. This figure has been completely redone in the manuscript. Orthogonality is now defined in terms of the median of all the angles between  $\nabla \gamma$ and  $\nabla \xi$  (or  $\nabla \xi'$ ). The results are depicted in the new Figure 2.

• 1149 : What would be the proportion of the world ocean covered in that range? To identify regions where a spiciness definition would be challenged could be an interesting add to the paper.

I have not attempted to quantify the accuracy of the quasi-linear approximation of density for the ocean's water masses, as it does not really matter for the arguments developed in the paper. My main aim was to construct a variable that can be used as a proxy for the spiciness variables of Jackett and McDougall (1995) and McDougall and Krzysik (2015) extending such variables to a wider range of reference pressures, allowing among other things to use a reference pressure  $p_r(S, \theta)$  or  $p_r(S_A, \Theta)$ .

- Along the manuscript, it is commonly referred to "orthogonality in physical space" and I think it would be nice to have a clear definition of what it means in introduction. I agree as I can see from the other comments that not doing so has created some confusion. This has now been fixed as described above.
- I have the feeling that in regions of the ocean with temperature-driven density, salinity anomalies will have be a better choice to construct an inert variable. Am I speculating too much? Would  $\Theta'$  be any better than  $S'_A$  where density is salinity-driven (eg, coastal ocean, near sea-ice, Mediterranean, Red, Black Seas, . . .)?

This is an interesting question, which I find difficult to answer. Indeed, as first showed by Jackett and McDougall (1995), all thermodynamic variables are approximately dynamically inert on a material approximately neutral  $\gamma = constant$  density surface. Whether the approximation is better for  $S'_A$  than  $\Theta'$ , and whether this can be proven, is an interesting suggestion that I am still not clear about.

**Response to Referee 2**

**General Comments** This manuscript aims to clarify the theoretical foundation for a spiciness variable sought by many oceanographers. The main idea is that, before considering spiciness, one must first construct a good neutral density variable that is materially conserved, and then most any materially conserved function can be used to construct a spiciness variable, simply by constructing its anomaly along neutral surfaces. The author also clarifies that pursuing orthogonality of spiciness and neutral density in  $S - \Theta$  space is misguided, and instead that the goal should be orthogonality of their gradients in physical space. Unfortunately, many of the advances of this paper are overstated, either lacking justification, detail, or novelty. There are several logical errors as well. These are discussed below. I believe this manuscript has the potential to nicely tie together the theory of spiciness variables, but Major Revisions are required to get there.

**Response and proposed changes** I thank the referee for his/her careful and comprehensive review, as well as for some thought provoking comments. In addition to addressing the numerous specific suggestions as detailed below, I have implemented the following main changes to address his/her most important comments.

- 1. To increase the novelty of the paper, I have redone all the calculations to be based on a new implementation of thermodynamic neutral density  $\gamma^T$  instead of  $\sigma_1$ . Outside the southern ocean, this variable is significantly more neutral than  $\sigma_1$  throughout the vertical column, and therefore a more satisfactory choice of quasi-neutral material density variable. I also showed how to construct potential spiciness and potential spiciness referenced to the non-constant reference pressure  $p_r(S, \theta)$  underlying the construction of  $\gamma$ .
- 2. I have rephrased the abstract to link the present work to the previous work by Jackett and McDougall (1985) as well as to McDougall and Giles (1987). I have also tried to make it clear in the paper that these two studies represent important precursors of the present work.
- 3. I have completely rewritten the parts about orthogonality in physical space and significantly toned down its usefulness or importance
- 4. I have significantly rewritten various parts that appear to have been causes of confusion.
- 5. After thinking about it, I think that I agree with the referee that the arguments presented only offer support for the pursuit of a globally-defined material density variable maximising neutrality, but that they fell short of representing first-principles arguments.

- 6. Section 4 provides an illustration of the results as before but has been completely rewritten to account for the remarks of the referees as well as to account for the use of  $\gamma^T$  instead of  $\sigma_1$ .
- 7. Many sections have been rewritten/rephrased in the light of the new insights achieved since the original submission.

**Response to specific comments**

• One of the major points of the paper, that what matters for spiciness is actually the neutral density variable, was made by Jacket and McDougall (1985). The author has acknowledged this in some places, but a reader could easily get the impression that this idea owes to this manuscript. A stand-out example is in the abstract (line 5): stating "contrary to what is usually assumed" is unfair. Anyone who has read Jackett and McDougall (1985) would not assume this. This phrase should be removed, and a citation to Jackett and McDougall (1985) given in the abstract.

Since the referee emphasises rigour and accuracy throughout his/her review, it seems important to point out that my point is actually that what determines the dynamical inertness of spiciness is the degree of neutrality of  $\gamma$ , which is quite different from what the referee says. Moreover, last time I checked, the term 'neutral density' is nowhere mentioned in Jackett and McDougall (1985) (JM85). Rather, JM85 use the term 'isopycnal surfaces', stating at the beginning of their paper: "In this paper we approximate the isopycnal surfaces by surfaces of constant potential density which we call  $\rho$ ." Moreover, JM85 also state: "the variations of any variable, when measured along isopycnal surfaces, are dynamically passive [...]". Since such a statement is technically true only on surfaces of constant in-situ density, there is no logical reason why readers would conclude that by 'isopycnal surfaces' JM85 actually mean 'neutral surfaces', especially as neither neutral surfaces nor patched potential density surfaces where yet in widespread use at the time. Even when such surfaces started to become popular, I am not aware that they have never associated with the dynamical inertness of spiciness before my paper.

I don't understand how my "contrary to what is usually assumed" is unfair to JM85, since what such a statement criticises is past studies contending that orthogonality and dynamical inertness are somehow connected, e.g., Veronis (1972). This is obviously not the case of JM85, which my paper praises for being the first to recognise that orthogonality is not connected with dynamical inertness. This being said, I agree that JM85 deserve a place in the abstract, not for promoting the use of neutral density, but for being the first to recognise the lack of connection between orthogonality and dynamical inertness, as well as for being the first to propose describing spiciness in terms of the isopycnal anomaly of some thermodynamic variable. As regards to the latter point, I don't think I give enough credit to JM85, which I hope has been

satisfactorily corrected in the revision of my paper.

As a final remark, I'd like to stress that I am not claiming credit for the idea that spiciness should be used in conjunction with neutral density, because there is no ambiguity that this is implied explicitly or implicitly in most if not all spicity/spiciness papers written post JM85, as is evident in Huang (2011) or McDougall and Krzysik (2015) for instance. I only claim credit for spelling a logical and rigorous argument establishing the link between dynamical inertness and neutrality, which had never been explicitly stated before, as far as I know.

• I find Fig 11 the most interesting aspect of this work. It is essentially a global test of the Jackett and McDougall (1985) idea, repeated here, that it is the anomaly  $\xi'$  is dynamically inert. The author's anomaly is defined as relative to a global isopycnal average. The results are evidently meaningful, but it is not entirely clear that more refined results could be obtained by refining the averaging procedure. McDougall and Giles (1987) argued in favor of studying property (salinity) anomalies relative to a local isopycnal average. To study a particular water mass intrusion, the state of the ocean far away should be irrelevant. It would therefore be prudent of the author to discuss the utility of using global isopycnal averages, and to locate the present work relative to the earlier work of McDougall and Giles (1987).

I fully agree with the referee and thank him/her for pointing out the highly relevant study by McDougall and Giles (1987), which I was not aware of. I have only touched upon global versus local definitions of  $\xi_r(\gamma)$  as I haven't really explored the issue thoroughly enough yet.

• Moreover, it would be interesting to add another panel to Fig 11 that tests the anomaly of a state variable that is specifically designed to be quite poor as spiciness-as-as-astate variable — but nonetheless may appear comparably good as spiciness-as-a-property (anomaly).

Conservative Temperature appears to be quite poor as a spiciness-as-a-state-function, and performs much better as an anomaly. It has now been added to the discussion.

• In addition to the question of which geographic data should enter the construction of the  $\xi_r(\gamma)$  function, the question of how this data is used must also be asked. Early on in the paper, the author describes this as the isopycnal mean, which presumably implies an arithmetic mean (this should be clarified). However, Section 4 seems to make this more general, stating only that  $\xi_r(\gamma)$  is a "suitably constructed function of density only". Should we use an arithmetic mean? If so, why? If we define  $\xi_r$  as the best such function, in some kind of a least-squares sense, would we discover that it is an arithmetic mean? Fig 8 provides a trivial example where  $\gamma$  and  $\xi$  are linear functions of space. Obviously, the real ocean presents a far more nonlinear problem, for there will not be a suitable function  $\xi_r(\gamma)$  that renders  $\nabla \xi'$  orthogonal to  $\nabla \gamma$ . Unless this general issue can be addressed, Section 4.1 is not of great theoretical or practical interest.

After further testing and thinking about the issue, I am retracting my earlier response because I am now believe that trying to maximise the orthogonality of  $\xi'$  to  $\gamma$  in physical space is not the right way to approach the construction of  $\xi_r(\gamma)$ . In truth, I don't know what is the best way to construct  $\xi_r(\gamma)$  and therefore decided to leave the issue open. In my paper, I chose a second order polynomial descriptor for  $\xi_r(\gamma)$ in order to ensure the smoothness and differentiability of the function, which is not usually guaranteed if a standard isopycnal average is used. As far as I can see, my choice appears to result in  $\xi'$  that appear to succeed well as water masses' indicators. Moreover, two of the resulting variables, namely  $\tau'_{ref}$  and  $S'_A$ , appear to have relatively small nonlinearities in  $S_A$  and  $\Theta$ , which suggests that they might be sufficiently conservative in practice (which means that they can be expected to mix linearly to a good approximation). I can only hope at this stage that my paper will stimulate further research on the topic and that others may succeed in coming up with a better way of doing things.

• Section 3 provides one way (among many) to nonlinearly scale the  $S - \Theta$  diagram so that both axes have common units [density], such that there is a well-defined spiciness variable  $\tau_{\ddagger}$  that is orthogonal to density on this diagram. However,  $\tau_{\ddagger}$  is subsequently dropped from the manuscript. It is claimed (line 168) that  $\tau_{\ddagger}$  is similar to  $\tau_{jmd}$ , but this is not proven or shown numerically. This manuscript would be considerably stronger if  $\tau_{\ddagger}$  were tested in Section 4.2 and shown to have some advantage over other spiciness/ spicity variables (including  $\tau_{jmd}$ , which it may well turn out to be very similar to). Otherwise, Section 3 seems to be of limited utility.

In the revised version of my paper,  $\tau_{jmd}$  and  $\tau_{\ddagger}$  are compared in Figs. 7 and 8, which appears to be sufficient to establish their similarity in  $(S_A, \Theta)$  space. The main advantage of  $\tau_{\ddagger}$  is its mathematically explicit character and continuous dependence on pressure, which makes it possible to construct the potential spiciness  $\tau_{ref}$  referenced to the variable reference pressure  $p_r(S, \theta)$  underlying the construction of thermodynamic neutral density. In the revised paper, Fig. 9 further establishes the similarity between the two variables.

Physically, I don't expect  $\tau_{\ddagger}$  to have many advantages over  $\tau_{jmd}$  for the study of spiciness anomalies. Its main advantages are practical ones, due to the Jacobian being non-zero everywhere in  $(S_A, \Theta)$  space, and due to its dependence on adjustable parameters being more explicit.

• The theoretical argument, opening Section 3, reaches the conclusion that the  $S - \Theta$  axes should be rescaled to have density as their common units, but this is commonly known. Huang et al (2018) pursues this, for example. The author does not make it clear why this rescaling of the  $S - \Theta$  diagram is superior to other rescalings, even linear ones.

I have decided to remove this part of the paper as I think that it was confusing and

not really necessary.

• In terms of structure, Section 4.2 "Illustrations" is more of a "Results" section, and does not fit well with the theoretical Section 4.1. I recommend splitting Section 4 into two sections, and expanding both, as described above.

I have completely rewritten Section 4, which is now entirely a 'results' section.

• The author claims that the anomaly  $\xi'$  "is the variable optimally suited for characterising ocean water masses" (line 4-5). However, this is not proven, nor is there any discussion about how such optimality would be measured. Claims of optimality appear in several other places in the manuscript. I recommend this loose language be qualified and proved, or else changed.

The term optimal now only appears in the abstract as part of the stated goal of the paper and has been removed from the rest of the paper. Optimal is only used in the sense of 'best', not in the sense of satisfying any particular metric. I think that this is common usage. It is proven in the sense that  $\xi'$  is demonstrably better than  $\xi$  for the purpose.

• Another one of the major results claimed is that this paper presents a "rigorous and first-principles theoretical justification for... a globally-defined material density variable  $\gamma(S, \theta)$  maximising neutrality" (e.g. lines 10, 115-116, 251-252). However, this justification is predicated upon the desire of oceanographers to have a spiciness variable. Though such a variable may be useful to possess, it does not itself have a rigorous and first-principles theoretical foundation, and so cannot be leveraged to justify such a  $\gamma$ .

It is not true that the theoretical justification for  $\gamma$  is "predicated upon the desire of oceanographers to have a spiciness variable". Indeed, the justification is predicated on the recognition that because in-situ density surfaces are inconvenient to use due to their strong dependence on pressure, there is a need for a material variable that for all practical purposes capture most of the dynamical features of in-situ density. This need is independent of the construction of a spiciness variable. Maximising neutrality is what maximise the ability of a material variable to behave like in-situ density for practical purposes. One such application is defining density-compensated  $(S, \theta)$  anomalies that are as passive as possible, but others exist: predicting thermal wind, static stability and so on...

**Specific Comments**

• 29: Another citation for thermobaric instability would be apt, here, such as Ingersoll (2005; JPO).

I have now cited this reference. Thanks for reminding me about it.

• 71: Some additional conditions are necessary to make this example true. As counterexample, take  $\gamma(S, \theta)$  and  $\xi(S, \theta)$  as constants: both are material, but the given

d does not satisfy property 2, since two distinct points  $(S_1, \theta_1)$  and  $(S_2, \theta_2)$  would nonetheless have d = 0.

I think the counterexample is not in the spirit of the paper. I have rephrased using the term 'nontrivial'  $\xi$  and  $\gamma$ .

• 80: Please provide further detail on the derivation of  $\gamma_S S' + \gamma_\theta \theta' \approx 0$ . Is one supposed to take the gradient of  $\gamma(S - S', \theta - \theta') = \gamma_0$  in the neutral tangent plane, and assume that  $\gamma$  is an approximately neutral density variable? This would lead to  $\gamma_S \nabla_n S' + \gamma_\theta \nabla_n \theta' \approx 0$ , but this differs from the stated equation by the presence of gradients. It is not clear whether the condiiton  $\gamma(S_r(\gamma_0), \theta_r(\gamma_0)) = \gamma_0$  is necessary "for all  $\gamma_0$ ", or just the  $\gamma_0$  under consideration. The equation is simply obtained from a Taylor series expansion around the reference

The equation is simply obtained from a Taylor series expansion around the reference  $S_r(\gamma_0)$  and  $\theta_r(\gamma_0)$  values, i.e.,  $\gamma(S_r(\gamma_0) + S', \theta_r(\gamma) + \theta') \approx \gamma(S_r(\gamma_0), \theta_r(\gamma_0)) + \gamma_S S' + \gamma_\theta \theta' + \cdots = 0$ , which implies  $\gamma_S S' + \gamma_\theta \theta' \approx 0$  at leading order.

• 93-94: Fig 2 does not show, as stated, that "the ability of a variable to characterise water masses is proportional to the degree of orthogonality between  $\nabla \xi$  and  $\nabla \gamma$  ..." It simply shows that the spatial gradient of different candidate spiciness variables make different angles with  $\nabla \gamma$ . . Fig 2 can only be interpreted as the author desires by referencing the interpretation of Fig. 1, that SA is a better spiciness variable than the other two. Even still, this is merely an interpretation or a "suggestion" at this stage.

I completely rewrote this part in the revision, hopefully in a more satisfactory way.

- Eq. (1): Please provide some details on the derivation of this equation. Tailleux (2016a) also lacks such details. This equation is obtained by the Jacobi method. It is explained in much details in Appendix A of Feistel (2018): Thermodynamic properties of seawater, ice and humid air: TEOS-10, before and beyond. Ocean Sciences, 14, 471–502. https://os.copernicus.org/articles/14/471/2018/os-14-471-2018.pdf
- Eq. (4):  $d_i$  needs to be defined. Also, it needs to be stated that this assumes is a perfectly neutral density variable, rather than "on any given density surface..." The expression is correct as stated.  $d_i$  is defined as the restriction of the total differential operator to an isopycnal surface  $\gamma = constant$ . The restriction depends on how  $\gamma$  is constructed. This has been clarified in the text.
- Eq. (6): Units error in the middle expression. X and Y have units of density, so cannot be added to the unitless value 1, which should be  $\rho_{00}$ . I agree. Thank for spotting this. Corrected.
- Eq. (11): This isn't really the total differential of  $\tau_{\ddagger}$  if it's at fixed pressure. But we can say that this is the total differential of potential spiciness. This has now been clarified

- 165:  $\tau$  has not been defined. All that can be said is that  $\tau_0$  is an arbitrary constant with units of density, and that  $\tau_{\ddagger}(S_0, \theta_0, p) = \tau_0)$ . I did not really understand this so did not act on it.
- 168:  $\tau_{\ddagger}$  is the exact solution to an approximate differential equation, but this does not mean  $\tau_{\ddagger}$  is an approximate solution of the exact differential equation. Here, Eq. (11) is the "approximate differential equation", which approximately matches (exactly in form, approximately in coefficients) with the "exact differential equation" set out by Jackett and McDougall (1985). If this logic were true, chaos (theory) would not exist. The referee has lost me. I have tried to rephrase the whole section, but am not sure that I have addressed this comment.
- Eq. (14): How did  $\rho_{00}$  become  $\rho_0$ ? I assume the neutral relation  $\nabla_i \theta = \alpha(S, \theta, p)\beta(S, \theta, p)^{-1}\nabla_i S$ was used, but this provides the second equality in (14) only if  $\rho_0 = \rho_{00}$ .  $\rho_0$  should be  $\rho_{00}$ , thanks for spotting it. Corrected.
- 196-7: Here, the author states that Section 3 showed spiciness can be theoretically justified to be orthogonal to density in thermohaline space, but elsewhere (e.g. line 208) stated that orthogonality in thermohaline space is "fundamentally ill-defined". This is confusing, to say the least. I remain unconvinced that Section 3 delivered what has been advertised here (line 196-7). Rather, Section 3 just showed that we can define an alternative, but only approximate, equation of state under which orthogonality in thermodynamic space is well-defined. This does not answer the theoretical questions surrounding spiciness in the real ocean.

After further thinking about the issue, I am withdrawing my previous response to this comment. Indeed, I have come to realising that orthogonality in physical space is not as useful as I initially thought. I have therefore completely rewritten all the relevant parts. Hopefully, the revision makes more sense and is less controversial.

• 204 and Eq. (15): This is introduced a bit sloppily. No definition is given for f so the reader is left to figure that out by understanding Eq. (15) and/or by comparison with  $\hat{\rho}$  earlier. Also,  $\partial \hat{f}/\partial p = \partial f/\partial p$  is used but not stated in Eq. (15), which would probably benefit by using the latter in the middle expression. Actually, since the same thing appears in Eq. (16), it may be better to simply provide an equation that does nothing more than define  $\hat{\nabla}$ , thereby eliminating these multi-part equations (15) and (16).

This bit has been removed in the revision.

- 206: "efficient" does not seem like the right word here. Maybe "compact"? I have removed this part altogether.
- 230: What is meant by "the values of  $\sigma_1$  contours retained in the nonlinear regression"? Is only some of the data shown in Fig 9 actually used in the nonlinear

regression that produces its red lines? And the data that is used has  $\sigma_1$  values between the largest and smallest of the thick black contours in Fig 10? The caption of Fig 10 helps support this interpretation, but even there it is confusing: the restricted range of  $\sigma_1$  used to compute the nonlinear regression should be defined by two  $\sigma_1$ values (a lower and upper bound) rather than four values (the thick contours). The relevant plot has been completely redrawn in the revision. Hopefully, it is now much clearer.

- 250: Jackett and McDougall (1985) should be cited here. Agreed
- 266-7: What would happen if you used a non-constant reference pressure for  $\tau_{\ddagger}$ , as suggested here? Actually, it's not clear what this even means: where does a reference pressure fit into  $\tau_{\ddagger}$ ?

The paper has been completely rewritten to now rely on the use of a non-constant reference pressure  $p_r(S,\theta)$ . This has led to the introduction of the potential spiciness variable  $\tau_{ref} = \tau_{\pm}(S,\theta,p_r(S,\theta))$  referenced to  $p_r(S,\theta)$ .

• 270: This claim, that Tailleux (2016b)'s density variable "maximizes neutrality while also being the only one that accounting for thermobaricity", is unfounded. Tailleux (2016b) only compared the neutrality of his density variable against a select few competitor density variables, namely two potential density variables,  $\gamma^n$  of Jackett and McDougall (1997), and a rational approximation of  $\gamma^n$  defined by McDougall and Jackett (2005; JMR). Conspicuously missing is the orthobaric density of de Szoeke et al (2000), not to mention the neutral density of Eden and Willebrand (1999). Moreover, since Tailleux (2016b)'s density variable was custom-built to mimic n of Jackett and Mc- Dougall (1997), and the latter exhibits better neutrality (Fig 6 of Tailleux (2016b)), it is unclear how the author can make this claim even if orthobaric density had been tested

Orthobaric density is not a purely material variable so does not fit in the present framework. However, McDougall and Jackett (2005)  $\gamma_a$  and Eden and Willebrand (1999)'s variable are presumably affected by thermobaricity on account to the way that they are constructed, so I agree with the referee that I was imprecise. The whole claim has been dropped in the revision.

• 275-6: The author has not shown that  $\xi'$  appears to be insensitive to the particular choice of  $\xi_r(\gamma)$ , since only one method for empirically constructing  $\xi_r(\gamma)$  was tested, namely the (arithmetic?) mean.

The use of the verb 'appears' makes my statement a subjective one. Nevertheless, I can try to be somewhat more precise in the revision.

• 278: Isn't  $\xi'$  conservative by definition? Since  $\xi$  and  $\gamma$  are assumed to be conservative throughout this manuscript, then  $\xi'$  should be too.

 $\gamma$  and  $\xi$  are assumed to be material and therefore Lagrangian invariants in the absence of diffusive sources/sinks of temperature and salinity, not conservative. The term conservative is used in the same sense as in McDougall (2003) paper on Conservative Temperature, i.e., as a variable satisfying an equation of the form

$$\frac{DH}{Dt} + \nabla \cdot \mathbf{F}_h = 0. \tag{1}$$

A priori,  $\gamma$  is not conservative since there are nonlinear production terms due to cabelling and thermobaricity.  $\xi$  will be similarly conservative depending on the magnitude of its linearities in S and  $\theta$ .

Fig 2: The source data should be restricted to be between, say, 500 and 1500 dbar, to remain near the reference pressure of σ1.
I have redone all the calculations using a new implementation of thermodynamic neutral density to avoid the issue altogether. The new Figure 4(b) shows that this variable is satisfactorily neutral throughout the water column outside the southern

ocean.

• Fig 2: The colors are a bit confusing. In the caption, spiciness and spicity are described as brown and orange, respectively – quite similar colors! This seems (to me) to describe more how they appear in the histogram when blended with other colors, not how they are in the legend.

This figure has been redone and redrawn completely and should be much clearer now

- Fig 9: It is nearly impossible to get much information from these panels. It is likely that most of what we see is due to outliers, and the vast majority of the data is lying on top of itself. Instead of a simple scatter plot, I suggest using a 2D histogram. I don't understand what would be the purpose of what the referee suggests. In any case, this figure has been modified to include data points from the global ocean as well. I suppose that this probably does not address the referee's comments, but I am not sure exactly what he/she has in mind.
- Fig 11: The colorbars all range between -2 and 2, but the units vary across panels. It would be better to let each colorbar cover the entire range of its variable, or perhaps to cover the variable's range up to two standard deviations, say. Dr. Zika suggested to normalise each variable by its standard deviation, which I did. The relative ranges of each variable is now much more comparable I think.
- Fig 11: Caption: Which contours of  $\sigma_1$  are shown in white? As said previously, all the calculations have been redone using a new implementation of thermodynamic neutral density. The contour labels for  $\gamma_{analytic}^T$  are now given in the new Figure 1 (panel d).

Technical Corrections Thank you for these. All these have been taken into account in the revision.

- 4-8: "The key results are:" should be "The key results are as follows." and each key result that follows should be a separate sentence. (What comes before a colon must be a complete sentence.)
- 9-10: Same issue as above.
- 19: behaves -¿ behave
- 28: sopycnal -¿ isopycnal
- 48 and 53: At this stage, it's unclear why or when "potential" should appear before "spicity" and "spiciness'.
- 50: remove "in general"
- 56, 277, 278: question mark should be a period, or rephrase so that a question is actually asked, rather than stating what the question is.
- 61: signal -¿ signals
- 67: The statement "checked in any good mathematics textbook" is rather cavalier, and would be better omitted. Simply naming the mathematical object d as a metric is enough.
- 69: Using "1" and "2" to identify data leads to the unfortunate notation of d(1; 2). I'd suggest using A and B instead of numbers.
- 72: The definition of fi is quite confusingly written, since (γ, ξ) is really meant to say γ or ξ"
- 105: This is usually called the "dianeutral vector" not the "neutral vector".
- 120: join  $\rightarrow$  joint
- 125: typo in the inline equation: the first S should be  $\theta$
- 125: J has already been defined and does not need to be stated again.
- Eq (14) and line 191:  $\tau$  should be  $\tau_{\ddagger}$ .
- 192: brackets -¿ braces in Eq. (14).
- 207: tilde is placed incorrectly, should be over  $\nabla$
- 259: all -; are all

- 260: "the one used in this study": it's unclear what "one" is referring to, since four candidates were tested, and the author's own variable was also presented.
- 263: "as the ... variable" -¿ "as ... variables"
- 264: mimic -¿ mimics
- Fig 2: The x axis label is missing two gradient symbols, in front of  $\sigma_1$  and  $\xi$ . Also, "11" - $\xi$  "1". Also, "less" - $\xi$  "least".
- Fig 9: "Fig. 11" -i" Fig. 1". Also, shouldn't "spiciness" and "spicity" be changed to "potential spiciness" and "potential spicity" throughout this caption? Also, the subscript for  $\tau_{\ddagger}$  is sideways on the y-axis label of panel (a).
- various: showed -¿ shown

**Response to Referee 3**

Tailleux presents new ideas around spiciness in the ocean. I think this is a worthwhile paper with some interesting points being made. A number of the key conclusions don't seem well supported though. Some points are presented as self-evident, yet their justification seems far from obvious. Furthermore, some analysis lacks rigour. I feel these are largely matters of presentation and I expect I will be able to recommend publication after major revision.

**Response and proposed changes** I thank Dr. Zika for his careful review and useful suggestions. After thinking more about his comments and doing some more calculations and analysis, I have come to realise that part of my analysis was indeed not sufficiently rigorous, especially the part related to orthogonality in physical space, which I now regard as not as useful as originally thought, although still relevant. The main changes implemented in response to his suggestions are:

- 1. I have completely rewritten the discussion of orthogonality in physical space and considerably toned down its significance and usefulness;
- 2. I have improved Fig. 11 by rescaling the variables by their standard deviation as suggested

In addition to such changes, I have also implemented several other changes in response to all the other comments received, the main ones being:

- I have replaced  $\sigma_1$  by the more neutral thermodynamic neutral density  $\gamma^T$ , for which I have proposed a new and computationally simpler implementation that should facilitate the reproducibility of my results. Section 2 has been modified to include a description of the new variable.
- To work with  $\gamma^T$ , the potential spiciness/spicity variables need to be referenced to a variable reference pressure  $p_r(S, \theta)$ . How to do that is now the topic of Section 3.
- Section 4 provides an illustration of the results as before but has been completely rewritten to account for the remarks of the referees as well as to account for the use of  $\gamma^T$  instead of  $\sigma_1$ .
- Many sections have been rewritten/rephrased in the light of the new insights achieved since the original submission.

**Specific issues:**

1. **Orthogonality** Tailleux argues that the most appropriate spice variable should be orthogonal in geographical coordinates. I actually think this is a very important point

but words like orthogonal and optimal are used frequently without their implementation actually being globally orthogonal, nor evidently 'optimal' in any way. Firstly, the importance of orthogonality is introduced with "As is well known, the most efficient way to represent a vector is achieved by decomposing it in an orthogonal basis" This statement (and similar statements about orthogonality) should be made more precisely. For example, does the word 'efficient' have a precise meaning here? If we are to apply rigour to the idea of developing an orthogonal basis, surely there is a fundamental issue that the gradient of any spice variable can vanish on an isopycnal (and clearly the along-section isopycnal gradient of all the spice variables shown in figure 11 vanish at various locations). The problem Tailleux is dealing with is in three-dimensional space yet neutral density and spice offer only two basis vectors. This should be clarified with regard to the motivation to have an orthogonal basis since the basis developed is clearly incomplete. I suggest a severe tone down of the language of 'orthogonal coordinates' unless these issues are to be discussed carefully. Perhaps more crucially, it is unclear where and to what degree the modified spice variable eta' is actually orthogonal. How de we know if the reference profile  $\xi_r(\sigma_1)$  is 'suitably constructed'? Fig.11 uses a polynomial fit of  $\xi_r(\sigma_1)$  for a specific section for  $\xi_r$ . Doesn't this imply there is no perfect orthogonality anywhere? Why not choose  $\eta_r$ to be  $\eta$  at a specific latitude and longitude so at least local orthogonality is ensured? Or one could use the global isopycnal average of  $\eta$  Why not these other choices? More generally, there is no attempt to quantify how 'optimal' different methods for making eta' orthogonal are despite it the word optimal being used frequently throughout the paper.

After further thinking about Dr. Zika's remarks, I would like to withdraw my previous response to this comment because I now think that the importance of orthogonality in physical space was overstated in my originally submitted paper and that is usefulness for constraining the choice of  $\xi$  or  $\xi_r(\gamma)$  is not as clear as I thought. As a result, I have significantly toned down the importance of orthogonality in the revision of my paper. As it turns out,  $\xi'$  is not always more orthogonal to  $\gamma$  in physical space than  $\xi$ , and upon further consideration, there is no strong reason that it should necessarily be. I hope that the referee will find the revised version of the paper more satisfactory in this respect. I am grateful to the referee for challenging me on that point, as I now realise that I had not fully taken into account all aspects of the problem. Because I completely rewritten the relevant parts of the manuscript, I believe that most of the other problems have been eliminated as a result.

2. Fig 11. I think the variables shown in Fig. 11 are even closer than they appear. Both potential spiciness and spicity are in units of kg/m3 while  $\Theta$  is in °C and S is in g/kg. As a consequence  $\Theta - \Theta_r$  is saturated and  $S - S_r$  is poorly resolved by the colour scale. There seems to not be a fundamental reason to care about the units of any of these coordinates since their utility is primarily in tracing water masses. So, I strongly encourage the author to rescale the colour axes (e.g. by dividing each by 1 standard deviation) so the variations in each variable are highlighted rather than their absolute values. This will likely show that all four variables look very similar in terms of their relative variations.

This is a very good idea that I have implemented and that indeed appears to be sufficient to considerably reduce the inter-differences between the different  $\xi'$ .

- 3. I am not sure if I saw it mentioned but it would be nice to see it pointed out that if the equation of state is indeed linear then all four of the diagnostics shown in Fig.11 should be proportional (at least I am sure this is the case for  $\Theta$  and S). This is not mentioned in the text. I am not convinced that this is really necessary as the true equation of state is not linear.
- 4. General references to previous work There are a lot of instances where what is written in previous work is generalised. These need to be either removed or replaced with concrete examples. For example on line 120 it says "So far, studies that have pursued orthogonality: : :have taken for granted: : :". Unless complete knowledge of all such studies can be claimed, it would be more appropriate to just point out that this has happened in some studies and provide references.

I have kept this in mind in revising the paper. I have tried to make more specific statements and to avoid excessive generalisations.

Other comments and suggestions:

Thank for these. I have done my best to address these.

- 1. There were a large number of typos and a few terms left un-defined. I have had the paper proof-read and converted to Word to try to minimise these problems. I hope the revision is better in this respect.
- 2. Generally, it makes more sense to me that 'I' is used instead of 'we' since this is a sole author paper. This goes against my nature, but I gave it a try.
- 3. A lot of the mathematics was difficult to follow often because basic variables and notation were not defined.I hope the revision is better
- 4. Line 14: What is a 'binary fluid' I think that most oceanographers know what a binary fluid is.
- 5. Line 25: What is "de-compensate"

At the surface, a density-compensated temperature anomaly will be modified by airsea interactions, without necessarily modifying the associated density-compensated salinity anomaly. As a result, the modified temperature anomaly can no longer be density compensated, hence the term 'de-compensate'.

- 6. Line 28: "isopycnal" corrected. thanks.
- 7. Line 45: "As \*shown\* in this paper" Part removed from the paper so no longer relevant
- 8. Line 72: I think I understand that f can be either gamma or eta. But as written it looks like f maps from Theta and S into gamma and eta space (e.g. the author writes  $f = (\gamma, \eta)$ ). This whole paragraph could be expanded for clarity as it is important. I have tried to clarify notations
- 9. Line 80: What is  $\gamma_S$ ? The partial derivative of gamma with respect to S? Yes. I think it is clear enough from context
- 10. Line 102: "As shown by" or "As Tailleux (2016a) showed" Corrected. Thanks
- 11. Eq 2: Define  $\rho_p$  and  $\rho_\eta$  Done. Thanks.
- 12. Line 120: "in a join\*t\* system". Also its not clear what a 'joint system of physical units' is.
  This part has been removed so no longer relevant
- Eq (5): Why no brackets around what is being logged here? This does not seem necessary here, as there is no ambiguity as to what is being logged.
- 14. Line 139: Why  $\rho_{00}$  and not just  $\rho_0$ ? I tend to use  $\rho_0$  as a function of p or z, hence the choice of  $\rho_{00}$  to refer to a constant reference density.
- 15. Line 177: Define 'quasi-material' I have replaced quasi-material by material
- 16. Personal note: In our recent paper, Zika, J. D., J-B. Sallée, A. C. Naveira-Garabato, A. J. Watson, A. Meijers, M-J. Messias, B. King, 2020: Tracking the spread of a passive tracer through Southern Ocean water masses. Ocean Science., 16, 323–336, 2020, we attempted to construct a coordinate which was locally orthogonal to the along isopycnal direction and also materially conserved. The coordinate was essentially  $S - S_r$ . We chose  $S - S_r$  because it was simpler to define than spice. Fig. 11 of this paper suggests this was a reasonable choice. Our salinity anomaly variable was used to help understand the ispoycnal spreading of a passive tracer. There are likely other examples of work that benefited from, or would have benefitted from, such 'spicy' coordinates. I feel this paper would be better motivated if more references

were made to such studies.

I now remember that Dr. Zika mentioned this study to me at Ocean Sciences, and I am sorry that I forgot to cite it. This has been cited in the revision.

---

## Author Response (AR2)

List of further changes made in response to the Editor's request:

- I have implemented all the technical corrections suggested
- Note that the lines 312 – 332 of the new uploaded pdf were missing from the submitted manuscript due to my misuse of the textcolor command, which truncated part of the text. As far as I am aware, only the above lines were missing. This is why the equation number was missing in the appendix. Sorry for that.